# Truncated Matrix Power Iteration for Differentiable DAG Learning

**Zhen Zhang**[*,1], **Ignavier Ng**[*,2], **Dong Gong**[3], **Yuhang Liu**[1], **Ehsan M Abbasnejad**[1], **Mingming Gong**[4], **Kun Zhang**[2,5], and **Javen Qinfeng Shi**[1]

[1]*The University of Adelaide*    [2]*Carnegie Mellon University*
[3]*The University of New South Wales*    [4]*The University of Melbourne*
[5]*Mohamed bin Zayed University of Artificial Intelligence*
*{zhen.zhang02,yuhang.liu01,ehsan.abbasnejad,javen.shi}@adelaide.edu.au*
*{ignavierng,kunz1}@cmu.edu    dong.gong@unsw.edu.au    mingming.gong@unimelb.edu.au*

## Abstract

Recovering underlying Directed Acyclic Graph (DAG) structures from observational data is highly challenging due to the combinatorial nature of the DAG-constrained optimization problem. Recently, DAG learning has been cast as a continuous optimization problem by characterizing the DAG constraint as a smooth equality one, generally based on polynomials over adjacency matrices. Existing methods place very small coefficients on high-order polynomial terms for stabilization, since they argue that large coefficients on the higher-order terms are harmful due to numeric exploding. On the contrary, we discover that large coefficients on higher-order terms are beneficial for DAG learning, when the spectral radiuses of the adjacency matrices are small, and that larger coefficients for higher-order terms can approximate the DAG constraints much better than the small counterparts. Based on this, we propose a novel DAG learning method with efficient truncated matrix power iteration to approximate geometric series based DAG constraints. Empirically, our DAG learning method outperforms the previous state-of-the-arts in various settings, often by a factor of 3 or more in terms of structural Hamming distance.

## 1 Introduction

Recovery of Directed Acyclic Graphs (DAGs) from observational data is a classical problem in many fields, including bioinformatics [31, 49], machine learning [19], and causal inference [39]. The graphical model produced by a DAG learning algorithm allows one to decompose the joint distribution over the variables of interest in a compact and flexible manner, and under certain assumptions [27, 39], these graphical models can have causal interpretations.

DAG learning methods can be roughly categorized into constraint-based and score-based methods. Most constraint-based approaches, *e.g.*, PC [37], FCI [11, 38], rely on conditional independence test and thus may require a large sample size [33, 43]. The score-based approaches, including exact methods based on dynamic programming [18, 35, 36], A* search [47, 48], and integer programming [13], as well as greedy methods like GES [9], model the validity of a graph according to some score function and are often formulated and solved as a discrete optimization problem. A key challenge for score-based methods is the combinatorial search space of DAGs [8, 10].

There is a recent interest in developing continuous optimization methods for DAG learning in the past few years. In particular, Zheng et al. [50] develops an algebraic characterization of DAG constraint

---

[*]Equal Contribution

36th Conference on Neural Information Processing Systems (NeurIPS 2022).

based on a continuous function, specifically matrix exponential, of the weighted adjacency matrix of a graph, and applies it to estimate linear DAGs with equal noise variances using continuous constrained optimization [4, 5]. The resulting algorithm is called NOTEARS. Yu et al. [45] proposes an alternative DAG constraint based on powers of a binomial to improve practical convenience. Wei et al. [44] unifies these two DAG constraints by writing them in terms of a general polynomial one. These DAG constraints have been applied to more complex settings, *e.g.*, with nonlinear relationships [20, 25, 45, 51], time series [26], confounders [6], and flexible score functions [53], and have been shown to produce competitive performance. Apart from the polynomial based constraints, Lee et al. [21], Zhu et al. [52] present alternative formulations of the DAG constraints based on the spectral radius of the weighted adjacency matrix. Instead of solving a constrained optimization problem, Yu et al. [46] proposes NOCURL which employs an algebraic representation of DAGs such that continuous unconstrained optimization can be carried out in the DAG space directly. However, the focus of these works [21, 46, 52] is to improve the scalability, and the performance of the resulting methods may degrade.

To avoid potential numeric exploding that may be caused by high-order terms in the polynomial-based DAG constraints, previous works use very small coefficients for high-order terms in DAG constraints [45, 50]. More precisely, the coefficients decrease quickly to very small values for the relatively higher order terms that may contain useful higher-order information. This may cause difficulties for the DAG constraints to capture the higher-order information for enforcing acyclicity. With such DAG constraints, a possible solution to maintain higher-order information for acyclicity is to apply an even larger penalty weight [45, 50], which, however, causes ill-conditioning issues as demonstrated by Ng et al. [24]. Moreover, these small coefficients will be multiplied on the gradients of higher-order terms during the optimization process, leading to gradient vanishing. In this paper, we show that, on the contrary, larger coefficients may be safely used to construct more informative higher-order terms without introducing numeric exploding. The reason is that the weighted adjacency matrix of a DAG must be nilpotent; thus the candidate adjacency matrix often has a very small spectral radius. As a result, the higher-order power of the matrix may be very close to zero matrix, and larger coefficients may be safely used without causing numeric exploding. We thus propose to use geometric series-based DAG constraint without small coefficients for more accurate and robust DAG learning. To relieve the computational burden from the geometric series, we propose an efficient Truncated Matrix Power Iteration (TMPI) algorithm with a theoretically guaranteed error bound. We summarize our contributions as the following:

- We demonstrate that the key challenge for DAG learning is the gradient vanishing issue instead of gradient and numeric exploding. For sparse graphs, the gradient vanishing issue can be severe.

- We design a DAG constraint based on finite geometric series which is an order-$d$ polynomial over adjacency matrices to escape from gradient vanishing. We show that the relatively large coefficients on the higher-order terms would not result in gradient exploding in practice.

- We show that there exists some constant $k \leqslant d$, *s.t.* our order-$d$ polynomial constraint can be reduced to order-$k$ polynomial without expanding its feasible set. Though finding such value of $k$ requires solving the NP-hard longest simple path problem, we develop a simple heuristic to find such $k$ with bounded error to the exact, order-$d$, DAG constraint. Based on this result, we propose a DAG constraint based on efficient Truncated Matrix Power Iteration (TMPI), and conduct experiments to demonstrate that the resulting DAG learning method outperforms previous methods by a large margin.

- We provide a systematical empirical comparison of the existing DAG constraints, including the polynomial and spectral radius based constraints, as well as the algebraic DAG representation of NOCURL.

## 2   Preliminaries

**DAG Model and Linear SEM**   Given a DAG model (a.k.a. Bayesian network) defined over random vector $\mathbf{x} = [x_1, x_2, \ldots, x_d]^\top$ and DAG $\mathcal{G}$, the distribution $P(\mathbf{x})$ and DAG $\mathcal{G}$ are assumed to satisfy the Markov assumption [27, 39]. We say that $\mathbf{x}$ follows a linear Structural Equation Model (SEM) if

$$\mathbf{x} = \mathbf{B}^\top \mathbf{x} + \mathbf{e}, \tag{1}$$

where $\mathbf{B}$ is the weighted adjacency matrix representing the DAG $\mathcal{G}$, and $\mathbf{e} = [e_1, e_2, \ldots, e_d]^\top$ denotes the exogenous noise vector consisting of $d$ independent random variables. With slight abuse of notation, we denote by $\mathcal{G}(\mathbf{B})$ the graph induced by weighted adjacency matrix $\mathbf{B}$. Moreover, we do not distinguish between random variables and vertices or nodes, and use these terms interchangeably.

Our goal is to estimate the DAG $\mathcal{G}$ from $n$ i.i.d. samples of $\mathbf{x}$, indicated by the matrix $\mathbf{X} \in \mathbb{R}^{n \times d}$. In general, the DAG $\mathcal{G}$ can only be identified up to its Markov equivalence class under the faithfulness [39] or sparsest Markov representation [30] assumption. It has been shown that for linear SEMs with homoscedastic errors, from which the noise terms are specified up to a constant [22], and for linear non-Gaussian SEMs, from which no more than one of the noise term is Gaussian [34], the true DAG can be fully identified. In this work, we focus on the linear SEMs with equal noise variances [28], which can be viewed as an instance of the former.

**DAG Constraints for Differentiable DAG Learning**  Recently, Zheng et al. [50] reformulated the DAG learning problem as the following continuous optimization problem with convex least-squares objective but non-convex constraint,

$$\min_{\mathbf{B} \in \mathbb{R}^{d \times d}} \| \mathbf{X} - \mathbf{X} \mathbf{B} \|_F^2 + \eta \| \mathbf{B} \|_1, \quad \text{subject to } h_{\exp}(\mathbf{B} \odot \mathbf{B}) = 0, \tag{2}$$

where $\| \cdot \|_F$ and $\| \cdot \|_1$ denote the Frobenius norm and elementwise $\ell_1$ norm, respectively, $\eta > 0$ is the regularization coefficient that controls the sparsity of $\mathbf{B}$, and $\odot$ denotes the Hadamard product, which is used to map $\mathbf{B}$ to a positive weighted adjacency matrix with the same structure. Given any $\tilde{\mathbf{B}} \in \mathbb{R}_{\geqslant 0}^{d \times d}$, the function $h_{\exp}$ is defined as

$$h_{\exp}(\tilde{\mathbf{B}}) = \mathrm{tr}(e^{\tilde{\mathbf{B}}}) - d = \mathrm{tr}\left( \sum_{i=1}^{\infty} \frac{1}{i!} \tilde{\mathbf{B}}^i \right), \tag{3}$$

and can be efficiently computed using various approaches [1, 2]. The key here is that $h_{\exp}(\tilde{\mathbf{B}}) = 0$ if and only if $\mathcal{G}(\tilde{\mathbf{B}})$ is a DAG. The formulation (2) involves a hard constraint, and Zheng et al. [50] adopted the augmented Lagrangian method [4, 5] to solve the constrained optimization problem. Ng et al. [24] demonstrated that the augmented Lagrangian method here behaves similarly to the classical quadratic penalty method [15, 29] in practice, and showed that it is guaranteed to return a DAG under certain conditions.

To improve practical convenience, Yu et al. [45] proposed a similar DAG constraint using an order-$d$ polynomial based on powers of a binomial, given by

$$h_{\mathrm{bin}}(\tilde{\mathbf{B}}) = \mathrm{tr}\left( \mathbb{I} + \alpha \tilde{\mathbf{B}} \right)^d - d = \mathrm{tr}\left( \sum_{i=1}^{d} \binom{d}{i} \alpha^i \tilde{\mathbf{B}}^i \right), \tag{4}$$

where $\alpha > 0$ is a hyperparameter and is often set to $1/d$, *e.g.*, in the implementation of DAG-GNN [45]. This function can be evaluated using $O(\log d)$ matrix multiplications using the exponentiation by squaring algorithm. Both $h_{\exp}(\tilde{\mathbf{B}})$ and $h_{\mathrm{bin}}(\tilde{\mathbf{B}})$ are formulated by Wei et al. [44] as a unified one

$$h_{\mathrm{poly}}(\tilde{\mathbf{B}}) = \mathrm{tr}\left( \sum_{i=1}^{d} c_i \tilde{\mathbf{B}}^i \right), \quad c_i > 0, \quad i = 1, 2, \ldots, d. \tag{5}$$

## 3 DAG Learning with Truncated Matrix Power Iteration

In this section, we first show that gradient vanishing is one main issue for previous DAG constraints and propose a truncated geometric series based DAG constraints to escape from the issue in Section 3.1. Then in Section 3.2 the theoretical properties of our DAG constraints are analyzed. In Section 3.3 we provide an efficient algorithm for our DAG constraints. In Section 3.4, we discuss the relation between our algorithm and previous works.

### 3.1  Gradient Vanishing Issue and Truncated Geometric DAG Constraint

Theoretically, any positive values of $c_i$ suffice to make the DAG constraint term $h_{\mathrm{poly}}(\tilde{\mathbf{B}})$ in Eq. (5) a valid one, *i.e.*, it equals zero if and only if the graph $\mathcal{G}(\tilde{\mathbf{B}})$ is acyclic. Thus previously small

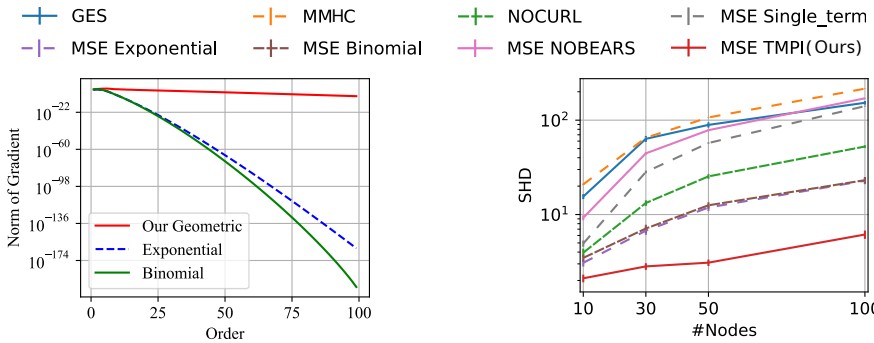

Figure 1: Previous polynomial based DAG constraints, including exponential (Eq. (3)) and binomial (Eq. (4) with $\alpha = 1/d$) constraints, suffer from gradient vanishing. **Left:** The norm of gradient on higher-order terms w.r.t. order on a 100-node graph generated using the ER2 setting in Section 4. Note that the gradients on different order terms refer to $\nabla_{\tilde{\mathbf{B}}} c_i \operatorname{tr}(\tilde{\mathbf{B}}^i)$, where, for different polynomial based DAG constraints, the coefficients $c_i$ are different. **Right:** We propose a geometric series based DAG constraint, named Truncated Matrix Power Iteration (TMPI), to escape from vanished gradients, which leads to a significant performance improvement in terms of Structural Hamming Distance (SHD) on ER2 graphs (see Section 4 for full experimental details and results).

coefficients for higher-order terms are preferred since they are known to have better resistance to numeric exploding of higher-order polynomials [45, 50]. These small coefficients forbid informative higher-order terms and cause severe gradient vanishing, as nilpotent properties of DAG result in candidate adjacency matrix with very small spectral radius (see empirical analysis in Figure 1 **left**). In this case, the weight updates during the optimization process can hardly utilize this higher-order information from the vanishingly small gradients. As a consequence, only lower-order information can be effectively captured, which may be detrimental to the quality of final solution.

To avoid the severe vanishing gradient issue in existing DAG constraints (*e.g.*, Eq. (3), (4)), we propose to discard the coefficients and use a finite geometric series based function:

$$h_{\text{geo}}(\tilde{\mathbf{B}}) = \operatorname{tr}\left(\sum_{i=1}^{d} \tilde{\mathbf{B}}^i\right), \tag{6a}$$

$$\nabla_{\tilde{\mathbf{B}}} h_{\text{geo}}(\tilde{\mathbf{B}}) = \left(\sum_{i=0}^{d-1}(i+1)\tilde{\mathbf{B}}^i\right)^{\top}. \tag{6b}$$

The gradient of $h_{\text{geo}}$ w.r.t. $\tilde{\mathbf{B}}$ denoted by $\nabla_{\tilde{\mathbf{B}}} h_{\text{geo}}(\tilde{\mathbf{B}})$ is an order-$(d-1)$ polynomial of $\tilde{\mathbf{B}}$.

Compared to (3) and (4), Eq. (6a) is a more precise DAG constraint escaped from from severe gradient vanishing. Also it is notable that due to the nilpotent properties of weighted adjacency matrices of DAGs, the higher-order terms in (6a) and (6b) often comes with small norms; thus, our DAG constraint may not suffer from numeric exploding in practice, based on the observations in Section 4.

In Figure 1 we compared the norm of gradients on higher-order terms of different DAG constraints, where we observe that for previous exponential-based (3) and binomial-based (4) DAG constraints, the gradients of higher-order terms converges to 0 very quickly as the order increases. Meanwhile, the gradient of higher-order terms in (6a) converges to 0 with a much slower speed and thus informative higher-order terms are preserved.

We further show that those non-informative higher-order terms that are close to zero in (6a) can be safely dropped to form a more efficient DAG constraint with some $k \leqslant d$, where we often have $k \ll d$ in practice:

$$h_{\text{trunc}}^k(\tilde{\mathbf{B}}) = \operatorname{tr}\left(\sum_{i=1}^{k} \tilde{\mathbf{B}}^i\right). \tag{7}$$

Naively, the DAG constraint (7) requires $\mathcal{O}(k)$ matrix multiplication which is far less than $\mathcal{O}(d)$ for (6a), and under mild conditions the error of (7) is tightly bounded (see Proposition 3 for details). In

Figure 1 **right** we show that our DAG constraint (7) (named TMPI) attains far better performance than previous ones as it escapes from gradient vanishing.

## 3.2 Theoretical Properties of Truncated Geometric DAG Constraint

We analyze the properties of the power of adjacency matrices, which play an important role in constructing DAG constraints. First notice that, for the $k^{\text{th}}$ power of an adjacency matrix $\tilde{\mathbf{B}} \in \mathbb{R}_{\geqslant 0}^{d \times d}$, its entry $(\tilde{\mathbf{B}}^k)_{ij}$ is nonzero if and only if there is a length-$k$ path (not necessarily simple) from node $X_i$ to node $X_j$ in graph $\mathcal{G}(\tilde{\mathbf{B}})$. This straightforwardly leads to the following property, which is a corollary of [44, Lemma 1].

**Proposition 1.** *Let $\tilde{\mathbf{B}} \in \mathbb{R}_{\geqslant 0}^{d \times d}$ be the weighted adjacency matrix of a graph $\mathcal{G}$ with $d$ vertices. $\mathcal{G}$ is a DAG if and only if $\tilde{\mathbf{B}}^d = \mathbf{0}$.*

Proposition 1 can be used to construct a single-term DAG constraint term, given by

$$h_{\text{single}}(\tilde{\mathbf{B}}) = \|\tilde{\mathbf{B}}^d\|_p, \tag{8}$$

where $\|\cdot\|_p$ denotes an elementwise $\ell_p$ norm. However, as the candidate adjacency matrix gets closer to acyclic, the spectral radius of $\tilde{\mathbf{B}}^d$ will become very small. As a result, the value $h_{\text{single}}(\tilde{\mathbf{B}})$, as well as its gradient $\nabla_{\tilde{\mathbf{B}}} h_{\text{single}}(\tilde{\mathbf{B}})$ may become too small to be represented in limited machine precision. This suggests the use of the polynomial based constraints for better performance.

Without any prior information of the graph, one has to compute the $d^{\text{th}}$ power of matrix $\tilde{\mathbf{B}}$, either using the DAG constraint terms in Eq. (8) or (5). However, if the length of the longest simple path in the graph is known, we can formulate a lower-order polynomial based DAG constraint.

**Proposition 2.** *Let $\tilde{\mathbf{B}} \in \mathbb{R}_{\geqslant 0}^{d \times d}$ be the weighted adjacency matrix of a graph $\mathcal{G}$ with $d$ vertices, and $k$ be the length of the longest simple path[1] in graph $\mathcal{G}$. The following three conditions are equivalent: (1) $\mathcal{G}$ is a DAG, (2) $\tilde{\mathbf{B}}^{k+1} = 0$, and (3) $h_{trunc}^k(\tilde{\mathbf{B}}) = 0$.*

The intuition of the above proposition is as follows. For the graph whose largest simple cycle has a length of $k$, it suffices to compute the $k^{\text{th}}$ power of its adjacency matrix, since computing any power larger than $k$ is simply traveling in the cycles repeatedly. Therefore, Proposition 2 suggests that $h_{\text{trunc}}^k(\tilde{\mathbf{B}}) = 0$ is a valid DAG constraint.

Unfortunately, the longest simple path problem for general graphs is known to be NP-hard [32], and thus we have to find an alternative way to seek the proper $k$. For a candidate adjacency matrix $\tilde{\mathbf{B}}$, a simple heuristic is to iterate over the power of matrices to find some $k$ *s.t.* matrix $\tilde{\mathbf{B}}^k$ is close enough to a zero matrix. Recall that Proposition 1 implies that the adjacency matrix of a DAG is nilpotent; therefore, when a candidate adjacency matrix $\tilde{\mathbf{B}}$ is close to being acyclic, its spectral radius would be close to zero, which implies that the $\tilde{\mathbf{B}}^k$ would converge quickly to zero as $k$ increases. Thus, if we use the above heuristic to find $k$, it is highly possible that we find some $k \ll d$.

By using an approximate $k$ instead of the exact one, the DAG constraint in Eq. (7) becomes an approximate one with the following bounds. Denoting by $\|\cdot\|_\infty$ the elementwise infinity norm (*i.e.*, maximum norm), we have the following proposition.

**Proposition 3.** *Given $\tilde{\mathbf{B}} \in \mathbb{R}_{\geqslant 0}^{d \times d}$, if there exists $k < d$ such that $\|\tilde{\mathbf{B}}^k\|_\infty \leqslant \epsilon < \frac{1}{(k+1)d}$, then we have*

$$0 \leqslant h_{geo}(\tilde{\mathbf{B}}) - h_{trunc}^k(\tilde{\mathbf{B}}) \leqslant \frac{1 - (d\epsilon)^{d/k-1}}{1 - (d\epsilon)^{1/k}} d^{2+1/k} \epsilon^{1+1/k},$$

$$0 \leqslant \|\nabla_{\tilde{\mathbf{B}}} h_{geo}(\tilde{\mathbf{B}}) - \nabla_{\tilde{\mathbf{B}}} h_{trunc}^k(\tilde{\mathbf{B}})\|_F \leqslant (k+1)d\epsilon \|\nabla_{\tilde{\mathbf{B}}} h_{geo}(\tilde{\mathbf{B}})\|_F.$$

**Remark 1.** *The condition $\epsilon < \frac{1}{(k+1)d}$ in the proposition above is such that the inequalities are non-vacuous, specifically since we must have $\|\nabla_{\tilde{\mathbf{B}}} h_{geo}(\tilde{\mathbf{B}}) - \nabla_{\tilde{\mathbf{B}}} h_{trunc}^k(\tilde{\mathbf{B}})\|_F / \|\nabla_{\tilde{\mathbf{B}}} h_{geo}(\tilde{\mathbf{B}})\|_F < 1$.*

---

[1] We use simple path to refer to a path that repeats no vertex, except that the first and last may be the same vertex [17, p. 363].

Proposition 3 provides an efficient way to bound the error of truncated constraints $h_{\text{trunc}}^k(\tilde{\mathbf{B}})$ to the geometric series based DAG constraints $h_{\text{geo}}(\tilde{\mathbf{B}})$. However, it also demonstrates a possible limitation for all polynomial based DAG constraints. As we know, when a solution is close to DAG, its spectral radius will be close to zero. In this situation, if the longest simple path in a graph is long, it is unavoidable to use high-order terms to capture this information while such high-order terms might be close to 0. In the worst case, the higher-order terms may underflow due to limited machine precision. Thus some specific numerically stable method may be required, and we leave this for future work.

### 3.3 Dynamic Truncated Matrix Power Iteration

Proposition 3 indicates that it may be safe to discard those matrix powers with index larger than $k$. In this case, the DAG constraint and its gradient correspond to order-$k$ and order-$(k-1)$ polynomials of $\tilde{\mathbf{B}}$, respectively. A straightforward way to compute the DAG constraint as well as the gradient is provided in Algorithm 1, which we call TMPI. The general polynomial based DAG constraint in (5) requires $\mathcal{O}(d)$ matrix multiplication, while Algorithm 1 reduces the number of multiplication to $\mathcal{O}(k)$. In practice, $k$ can be much smaller than $d$; for example, when recovering 100-node Erdős–Rényi DAG with expected degree 4, $k$ is often smaller than 20.

---

**Algorithm 1** Truncated Matrix Power Iteration (TMPI)

**Input:** $\tilde{\mathbf{B}} \in \mathbb{R}_{\geqslant 0}^{d \times d}, \epsilon$
**Output:** $h(\tilde{\mathbf{B}}), \nabla_{\tilde{\mathbf{B}}} h(\tilde{\mathbf{B}})$
1: $i \leftarrow 2, h(\tilde{\mathbf{B}}) \leftarrow \text{tr}(\tilde{\mathbf{B}}), \nabla_{\tilde{\mathbf{B}}} h(\tilde{\mathbf{B}}) \leftarrow \mathbb{I}$
2: **while** $i \leqslant d$ **do**
3: $\quad \nabla_{\tilde{\mathbf{B}}} h(\tilde{\mathbf{B}}) \leftarrow \nabla_{\tilde{\mathbf{B}}} h(\tilde{\mathbf{B}}) + i(\tilde{\mathbf{B}}^{i-1})^\top$
4: $\quad \tilde{\mathbf{B}}^i \leftarrow \tilde{\mathbf{B}}\tilde{\mathbf{B}}^{i-1}$
5: $\quad h(\tilde{\mathbf{B}}) \leftarrow h(\tilde{\mathbf{B}}) + \text{tr}(\tilde{\mathbf{B}}^i)$
6: $\quad$ **if** $\|\tilde{\mathbf{B}}^i\|_\infty \leqslant \epsilon$ **then break**
7: $\quad i \leftarrow i + 1$
8: **end while**
9: **Output** $h(\tilde{\mathbf{B}}), \nabla_{\tilde{\mathbf{B}}} h(\tilde{\mathbf{B}})$

---

**Algorithm 2** Fast TMPI

**Input:** $\tilde{\mathbf{B}} \in \mathbb{R}_{\geqslant 0}^{d \times d}, \epsilon$
**Output:** $h(\tilde{\mathbf{B}}), \nabla_{\tilde{\mathbf{B}}} h(\tilde{\mathbf{B}})$
1: $i \leftarrow 1$
2: $\tilde{\mathbf{B}}_f \leftarrow \tilde{\mathbf{B}}, \tilde{\mathbf{B}}_g \leftarrow \mathbb{I}, \tilde{\mathbf{B}}_p \leftarrow \tilde{\mathbf{B}}$ {$\tilde{\mathbf{B}}_f$ stores $f_i(\tilde{\mathbf{B}})$, $\tilde{\mathbf{B}}_g$ stores $\nabla_{\tilde{\mathbf{B}}} \text{tr}(f_i(\tilde{\mathbf{B}}))$, and $\tilde{\mathbf{B}}_p$ stores $\tilde{\mathbf{B}}^i$}
3: **while** $i \leqslant d$ **do**
4: $\quad \tilde{\mathbf{B}}_f^{old} \leftarrow \tilde{\mathbf{B}}_f, \tilde{\mathbf{B}}_g^{old} \leftarrow \tilde{\mathbf{B}}_g, \tilde{\mathbf{B}}_p^{old} \leftarrow \tilde{\mathbf{B}}_p$
5: $\quad \tilde{\mathbf{B}}_f \leftarrow \tilde{\mathbf{B}}_f^{old} + \tilde{\mathbf{B}}_p^{old}\tilde{\mathbf{B}}_f^{old}, \tilde{\mathbf{B}}_p \leftarrow \tilde{\mathbf{B}}_p^{old}\tilde{\mathbf{B}}_p^{old}$
6: $\quad \tilde{\mathbf{B}}_g \leftarrow \tilde{\mathbf{B}}_g^{old} + \tilde{\mathbf{B}}_p^{old\top}\tilde{\mathbf{B}}_g^{old} + i(\tilde{\mathbf{B}}_f - \tilde{\mathbf{B}}_f^{old} + \tilde{\mathbf{B}}_p^{old} - \tilde{\mathbf{B}}_p)$
7: $\quad$ **if** $\|\tilde{\mathbf{B}}_p\|_\infty \leqslant \epsilon$ **then break**
8: $\quad i \leftarrow 2i$
9: **end while**
10: **Output** $\text{tr}(\tilde{\mathbf{B}}_f), \tilde{\mathbf{B}}_g$

---

**Fast Truncated Matrix Power Iteration** We exploit the specific structure of geometric series to further accelerate the TMPI algorithm from $\mathcal{O}(k)$ to $\mathcal{O}(\log k)$ matrix multiplications. This is acheived by leveraging the following recurrence relations, with a proof provided in Appendix A.4.

**Proposition 4.** *Given any $d \times d$ real matrix $\tilde{\mathbf{B}}$, let $f_i(\tilde{\mathbf{B}}) = \tilde{\mathbf{B}} + \tilde{\mathbf{B}}^2 + \cdots + \tilde{\mathbf{B}}^i$ and $h_i(\tilde{\mathbf{B}}) = \text{tr}(f_i(\tilde{\mathbf{B}}))$. Then we have the following recurrence relations:*

$$f_{i+j}(\tilde{\mathbf{B}}) = f_i(\tilde{\mathbf{B}}) + \tilde{\mathbf{B}}^i f_j(\tilde{\mathbf{B}}),$$
$$\nabla_{\tilde{\mathbf{B}}} h_{i+j}(\tilde{\mathbf{B}}) = \nabla_{\tilde{\mathbf{B}}} h_i(\tilde{\mathbf{B}}) + (\tilde{\mathbf{B}}^i)^\top \nabla_{\tilde{\mathbf{B}}} h_j(\tilde{\mathbf{B}}) + i f_j(\tilde{\mathbf{B}})^\top (\tilde{\mathbf{B}}^{i-1})^\top.$$

This straightfowardly implies the following corollary, with a proof given in Appendix A.5.

**Corollary 1.** *Given any matrix $d \times d$ real matrix $\tilde{\mathbf{B}}$, let $f_i(\tilde{\mathbf{B}}) = \tilde{\mathbf{B}} + \tilde{\mathbf{B}}^2 + \cdots + \tilde{\mathbf{B}}^i$ and $h_i(\tilde{\mathbf{B}}) = \text{tr}(f_i(\tilde{\mathbf{B}}))$. Then we have have the following recurrence relations:*

$$f_{2i}(\tilde{\mathbf{B}}) = (\mathbb{I} + \tilde{\mathbf{B}}^i) f_i(\tilde{\mathbf{B}}),$$
$$\nabla_{\tilde{\mathbf{B}}} h_{2i}(\tilde{\mathbf{B}}) = \nabla_{\tilde{\mathbf{B}}} h_i(\tilde{\mathbf{B}}) + (\tilde{\mathbf{B}}^i)^\top \nabla_{\tilde{\mathbf{B}}} h_i(\tilde{\mathbf{B}}) + i \left( f_{2i}(\tilde{\mathbf{B}}) - f_i(\tilde{\mathbf{B}}) + \tilde{\mathbf{B}}^i - \tilde{\mathbf{B}}^{2i} \right)^\top.$$

Instead of increasing the index $i$ (*i.e.*, power of $\tilde{\mathbf{B}}$) arithmetically by 1, as in line 9 of Algorithm 1, Corollary 1 suggests using its recurrence relations to increase the index $i$ geometrically by a factor of 2, until the condition $\|\tilde{\mathbf{B}}^i\|_\infty \leqslant \epsilon$ is satisfied. The full procedure named Fast TMPI is described in Algorithm 2, which requires $\mathcal{O}(\log k)$ matrix multiplications and $\mathcal{O}(d^2)$ additional storage. An

example implementation is given in Listing 1 in Appendix B.1. Similar strategy can be applied to the original geometric series based constraint (6a) to form a fast algorithm that requires $\mathcal{O}(\log d)$ matrix multiplications. A comparison of the time complexity between our proposed constraint and the existing ones is provided in Appendix B.2.

**The Full Optimization Framework** For NOTEARS, we apply the same augmented Lagrangian framework [4, 5] adopted by Zheng et al. [50] to convert the constrained optimization problem into the following unconstrained problem

$$\min_{\mathbf{B} \in \mathbb{R}^{d \times d}} \quad \| \mathbf{X} - \mathbf{X} \mathbf{B} \|_F^2 + \eta \| \mathbf{B} \|_1 + \frac{\rho}{2} h_{\text{trunc}}^k (\mathbf{B} \odot \mathbf{B})^2 + \alpha h_{\text{trunc}}^k (\mathbf{B} \odot \mathbf{B}), \tag{9}$$

where the parameters $\rho$ and $\alpha$ are iteratively updated using the same strategy as Zheng et al. [50]. We also apply the proposed DAG constraint to GOLEM [23] that adopts likelihood-based objective with soft constraints, leading to the unconstrained optimization problem

$$\min_{\mathbf{B} \in \mathbb{R}^{d \times d}} \quad \frac{d}{2} \log \| \mathbf{X} - \mathbf{X} \mathbf{B} \|_F^2 - \log | \det(\mathbb{I} - \mathbf{B}) | + \eta_1 \| \mathbf{B} \|_1 + \eta_2 h_{\text{trunc}}^k (\mathbf{B} \odot \mathbf{B}), \tag{10}$$

where $\eta_1, \eta_2 > 0$ are the regularization coefficients. In both formulations above, we use Algorithm 1 or 2 to dynamically compute $h_{\text{trunc}}^k (\mathbf{B} \odot \mathbf{B})$ (and the value of $k$) during each optimization iteration based on a predefined tolerance $\epsilon$.

### 3.4 Relation to Previous Works

Our algorithm is closely related to the following DAG constraint considered by Zheng et al. [50].

**Lemma 1** (Zheng et al. [50, Proposition 1]). *Let $\tilde{\mathbf{B}} \in \mathbb{R}_{\geqslant 0}^{d \times d}$ with spectral radius smaller than one be the weighted adjacency matrix of a graph $\mathcal{G}$ with $d$ vertices. $\mathcal{G}$ is a DAG if and only if*

$$\text{tr}(\mathbb{I} - \tilde{\mathbf{B}})^{-1} = d. \tag{11}$$

When the spectral radius is smaller than one, the inverse of $\mathbb{I} - \tilde{\mathbf{B}}$ is simply the following infinite geometric series:

$$(\mathbb{I} - \tilde{\mathbf{B}})^{-1} = \sum_{i=0}^{\infty} \tilde{\mathbf{B}}^i. \tag{12}$$

Zheng et al. [50] argued that (12) is not well defined for $\tilde{\mathbf{B}}$ with spectral radius larger than 1, and in practice the series (12) may explode very quickly. Based on this argument, they adopted the matrix exponential for constructing the DAG constraint. For the same reason, Yu et al. [45] also used very small coefficients for higher-order terms in polynomial based DAG constraints. However, as we illustrate in Section 3, the DAG constraints proposed by Yu et al. [45], Zheng et al. [50] may suffer from gradient vanishing owing to the small coefficients for higher-order terms. In fact, our constraint (7) serves a good approximation for (11) since we are actually looking for an approximate inverse of $\mathbb{I} - \tilde{\mathbf{B}}$ with a bounded error.

**Proposition 5.** *Given $\tilde{\mathbf{B}} \in \mathbb{R}_{\geqslant 0}^{d \times d}$, if there exists $k$ such that $\|\tilde{\mathbf{B}}^k\|_\infty \leqslant \epsilon < 1/d$, then we have*

$$\left\| (\mathbb{I} - \tilde{\mathbf{B}})^{-1} - \sum_{i=0}^{k} \tilde{\mathbf{B}}^k \right\|_F \leqslant d\epsilon \left\| (\mathbb{I} - \tilde{\mathbf{B}})^{-1} \right\|_F.$$

Previous works [24, 44] pointed out that for DAG constraints with form $h(\mathbf{B} \odot \mathbf{B})$, the Hadamard product would lead to vanishing gradients in the sense that the gradient $\nabla_{\mathbf{B}} h(\mathbf{B} \odot \mathbf{B}) = 0$ if and only if $h(\mathbf{B} \odot \mathbf{B}) = 0$. Ng et al. [24] further showed that this may lead to ill-conditioning issue when solving the constrained optimization problem. One may replace the Hadamard product with absolute value, as in Wei et al. [44], but the non-smoothness of absolute value may make the optimization problem difficult to solve and lead to poorer performance [24]. As pointed out by Ng et al. [24], Quasi-Newton methods, *e.g.*, L-BFGS [7], still perform well even when faced with the ill-conditioning issue caused by vanishing gradients of constraints with form $h(\mathbf{B} \odot \mathbf{B})$. In this work, we identify another type of gradient vanishing issue caused by (1) the higher-order terms and small coefficients in function $h$ itself and (2) the small spectral radius of candidate matrix $\mathbf{B}$, and further provide an effective solution by constructing a Truncated Matrix Power Iteration based DAG constraint.

# 4 Experiments

We compare the performance of different DAG learning algorithms and DAG constraints to demonstrate the effectiveness of the proposed TMPI based DAG constraint. We provide the main experimental results in this section, and further empirical studies can be found in Appendix C.[2]

## 4.1 DAG Learning

We first conduct experiments on synthetic datasets using similar settings as previous works [23, 46, 50]. In summary, random Erdös-Rényi graphs [14] are generated with $d$ nodes and $kd$ expected edges (denoted by ER$k$). Edge weights generated from uniform distribution over the union of two intervals $[-2, -0.5] \cup [0.5, 2.0]$ are assigned to each edge to form a weighted adjacency matrix $\mathbf{B}$. Then $n$ samples are generated from the linear SEM $\mathbf{x} = \mathbf{B}\mathbf{x} + \mathbf{e}$ to form an $n \times d$ data matrix $\mathbf{X}$, where the noise $\mathbf{e}$ are i.i.d. sampled from Gaussian, Exponential, or Gumbel distribution. We set the sample size $n = 1000$ and consider 4 different number of nodes $d = 10, 30, 50, 100$ unless otherwise stated. For each setting, we conduct 100 random simulations to obtain an average performance.

**Methods and Baselines** We consider continuous DAG learning methods based on both Mean Squared Error (MSE) loss (see Eq. (9)) and likelihood-based (see Eq. (10)) loss from GOLEM [23]. For both losses, we include the following DAG constraints: (1) MSE/Likelihood loss+Exponential DAG constraint (3), *i.e.* NOTEARS [50]/GOLEM[23] ; (2) MSE/Likelihood loss+Binomial DAG constraint (4) from DAG-GNN [45]; (3) MSE/Likelihood loss+Spectral radius based DAG constraint from NOBEARS [21]; (4) MSE/Likelihood loss+Single Term DAG constraint (8); (5) MSE/Likelihood loss+Truncated Matrix Power Iteration (TMPI) based DAG constraint (7). For the NOCURL method [46], since it is a two stage methods where the initialization stage rely on MSE loss and Binomial DAG constraint, we include it as a baseline, but do not combine it with likelihood loss. The other baselines include GES [9] and MMHC [41, 42][3].

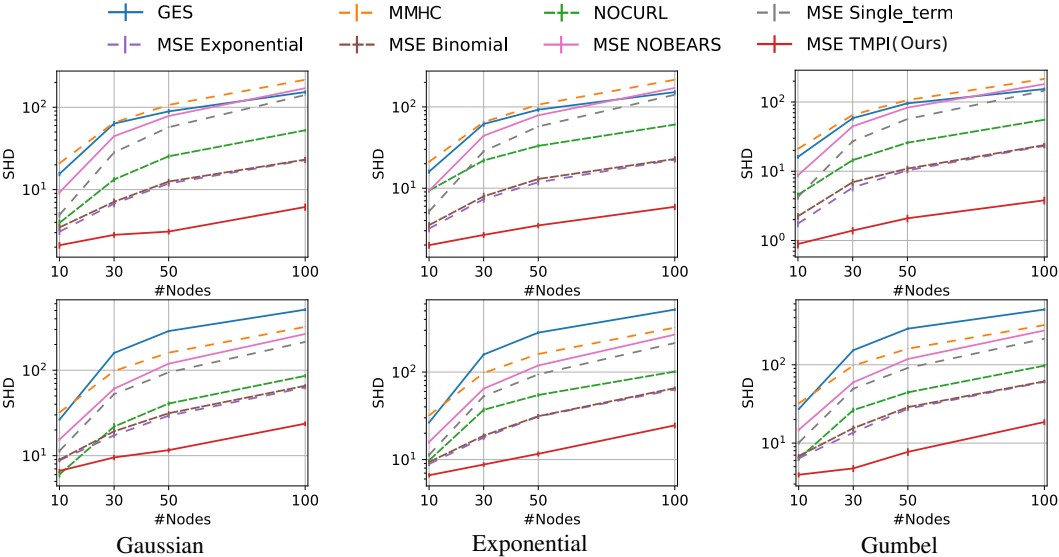

Figure 2: DAG learning results of MSE loss (*e.g.* Eq. (9)) based algorithms and baselines in terms of SHD (lower is better) on ER2 (**top**) and ER3 (**bottom**) graphs, where our algorithm consistently outperforms others almost by an order of magnitude. Error bars represent standard errors over 100 simulations.

For MSE loss based DAG learning methods (9), we set $\eta = 0.1$ and use the same strategy as Zheng et al. [50] to update the parameters $\rho$ and $\alpha$. For likelihood loss based DAG learning methods (10), we set $\eta_1 = 0.02$ and $\eta_2 = 5.0$ as Ng et al. [23]. For the Binomial DAG constraints (4), the parameter $\alpha$ is set to $1/d$ as Yu et al. [45]. For our TMPI DAG constraint, we set the parameter $\epsilon = 10^{-6}$. For the NOCURL method, we use their public available implementation and default

---

[2]The source code is avaiable at here.

[3]The implementation is provided here.

parameter setting[4]. For GES and MMPC, we use the implementation and default parameter setting in the `CausalDiscoveryToolbox` package [16].

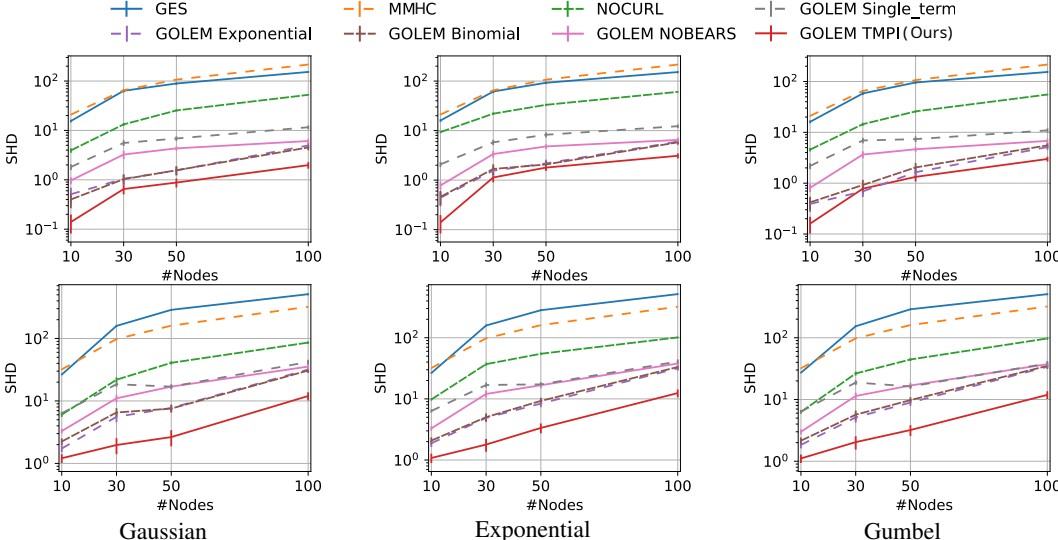

Figure 3: DAG learning results of likelihood loss (*i.e.* Eq. (10)) based algorithms and baselines in terms of SHD (lower is better) on ER2 (**top**) and ER3 (**bottom**) graphs, where our algorithm consistently outperforms others in a large margin in almost all cases. Error bars represent standard errors over 100 simulations.

**Results**    The results for MSE/likelihood based DAG learning algorithm are shown in Figure 2/3, where we compare the Structural Hamming Distance (SHD) of different algorithms. The complete results for other metrics are reported in Appendix C.1. The original NOTEARS (*i.e.* MSE+Exponential) / GOLEM (*i.e.* GOLEM+Exponential) has similar performance as MSE/GOLEM+Binomial. This may not be surprising because the coefficients of each order term in the Exponential and Binomial constraints are actually very close (see Figure 1 for an example). The performance of algorithms using single-term DAG constraint is often not comparable to the other polynomial-based constraints since it suffers more from gradient vanishing. The performance of algorithms using the spectral radius based constraints (BEARS) are also worse than the ones using Exponential and Binomial constraints. The algorithms using our TMPI constraints outperforms all others in almost all cases, which demonstrate the effectiveness of our DAG constraint.

**Additional Experiments**    We conduct additional experiments with scale-free graphs, smaller sample size (*i.e.*, $n = 200$ samples), larger graphs (*i.e.*, 500-node graphs), and denser graphs (*i.e.*, ER6 graphs). Due to the space limit, the results of these experiments are provided in Appendices C.2, C.3, C.4, and C.6, respectively, which further demonstrate the effectiveness of our proposed DAG constraint. We also compare our method with NOFEARS [44] in Appendix C.5.

### 4.2   Extension to Nonlinear Case

Although the description of our algorithm in Section 3 focuses on the linear case, it can be straightforwardly extended to the nonlinear case. In this section, we extend our DAG constraint to nonlinear case by replacing the binomial DAG constraint in DAG-GNN [45], NOTEARS-MLP [51] and GRAN-DAG [20] with our TMPI constraint.

**Synthetic Data**    We compare our algorithm and DAG-GNN on 50-node ER1 graph with nonlinear SEM $\mathbf{x} = \mathbf{B}\cos(\mathbf{x}+1) + \mathbf{e}$ over 5 simulations. Our algorithm achieves SHD of $22.2 \pm 4.2$ while DAG-GNN achieves SHD of $25.2 \pm 4.5$. We simulate a nonlinear dataset with 50-node ER1 graphs and each function being an MLP as Zheng et al. [51]. The original NOTEARS-MLP has SHD $16.9 \pm 1.5$, while NOTEARS-MLP with our TMPI constraint achieves an SHD of $14.9 \pm 1.3$.

---

[4]https://github.com/fishmoon1234/DAG-NoCurl/blob/master/main_efficient.py from commit 4e2a1f4 on 14 Jun 2021. Better performance may be attained by tuning the hyper-parameters, but the current best reported performance is slightly worse than NOTEARS in most settings.

**Real Data** We also run an experiment on the Protein Signaling Networks dataset [31] consisting of $n = 7466$ samples with $d = 11$ nodes. The dataset is widely used to evaluate the performance of DAG learning algorithms; however, since the identifiability of the DAG is not guaranteed, we compare both SHD and SHDC (SHD of CPDAG), following Lachapelle et al. [20]. In this experiment, we replace the original DAG constraints in DAG-GNN [45] and Gran-DAG [20] with our TMPI DAG constraints. The results are shown in Table 1, showing that by replacing the DAG constraint with our TMPI constraint, there is a performance improvement on SHD and SHDC for almost all cases.

## 4.3 Ablation Study

We use the ER3 dataset in previous section to conduct an ablation study to compare the performance of the Geometric constraint (6a) and our TMPI constraint (7), with both the naive implementation (*i.e.*, linear complexity) and fast implementation (*i.e.*, logarithmic complexity) from Algorithms 1 and 2, respectively. The results are shown in Figure 4. The SHD between Geometric constraint and TMPI constraint in both the naive and fast implementations are similar, and our Fast TMPI algorithm runs significantly faster than the other three, especially when the graph size is large. We also conduct further empirical study to illustrate the numeric difference between different $k$ for our DAG constraint; due to limited space, a detailed discussion is provided in Appendix C.7.

Table 1: Experimental results on Sachs dataset. Our TMPI DAG constraint consistently outperforms other DAG constraints.

|  | SHD | SHDC |
| --- | --- | --- |
| DAG-GNN | 16 | 21 |
| DAG-GNN/TMPI (Ours) | 16 | 17 |
| Gran-DAG | 13 | 11 |
| Gran-DAG/TMPI (Ours) | **12** | **9** |
| Gran-DAG++ | 13 | 10 |
| Gran-DAG++/TMPI (Ours) | **12** | **9** |

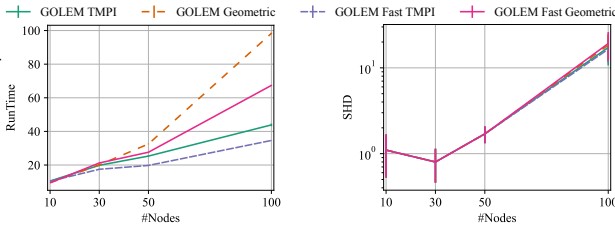

Figure 4: Comparison of Geometric and TMPI constraints, all algorithm achieves similar performance but our fast algorithm runs significantly faster. **Left:** Running time; **Right:** SHD. Error bars represent standard errors over 100 simulations.

## 5 Conclusion and Future Work

While polynomial constraints-based DAG learning algorithms achieve promising performance, they often suffer from gradient vanishing in practice. We identified one important source of gradient vanishing: improper small coefficients for higher-order terms, and proposed to use the geometric series based DAG constraints to escape from gradient vanishing. We further proposed an efficient algorithm that can evaluate the constraint with bounded error in $\mathcal{O}(\log k)$ complexity, where $k$ is often far smaller than the number of nodes. By replacing the previous DAG constraints with ours, the performance of DAG learning can often be significantly improved in various settings.

A clear next step would be finding strategies to adaptively adjust the coefficients for higher-order terms in polynomial based constraints. In a nutshell, polynomial-based constraints may explode on candidate graphs that are far from acyclic, and also suffer from gradient vanishing problem when the graph is nearly acyclic. Fixed coefficients are not able to solve both problems at the same time. Another possible direction is to use the sparse properties in DAGs to make DAG learning more efficient. Since the adjacency matrix of DAG must be nilpotent, there might exist more compact representation of the adjacency matrix, such as a low-rank CANDECOMP/PARAFAC (CP) decomposition, which can potentially be used to derive an efficient algorithm for very large graphs.

## Acknowledgements

We thank the anonymous reviewers, especially Reviewer iGwL, for their helpful comments. ZZ, DG, YL, EA and JQS were partially supported by Centre for Augmented Reasoning at the Australian Institute for Machine Learning. IN and KZ were partially supported by the National Institutes of Health (NIH) under Contract R01HL159805, by the NSF-Convergence Accelerator Track-D award #2134901, by a grant from Apple Inc., and by a grant from KDDI Research Inc.. MG was supported by ARC DE210101624.

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
