# Appendices

## A  Proofs

### A.1  Proof of Proposition 1

**Proposition 1.** *Let $\tilde{\mathbf{B}} \in \mathbb{R}_{\geqslant 0}^{d \times d}$ be the weighted adjacency matrix of a graph $\mathcal{G}$ with $d$ vertices. $\mathcal{G}$ is a DAG if and only if $\tilde{\mathbf{B}}^d = \mathbf{0}$.*

*Proof.* Any path in a $d$-node DAG cannot be longer than $d - 1$, and thus we must have $(\tilde{\mathbf{B}}^d)_{ij} = 0$ for all $i, j$, which indicates for DAG we must have $\tilde{\mathbf{B}}^d = \mathbf{0}$. On the other side, a nilpotent adjacency matrix must form a DAG [44, Lemma 1].  □

### A.2  Proof of Proposition 2

**Proposition 2.** *Let $\tilde{\mathbf{B}} \in \mathbb{R}_{\geqslant 0}^{d \times d}$ be the weighted adjacency matrix of a graph $\mathcal{G}$ with $d$ vertices, let $k$ be the length of longest simple path in $\mathcal{G}$, the following 3 conditions are equivalent: (1) $\mathcal{G}$ is a DAG, (2) $\tilde{\mathbf{B}}^{k+1} = 0$, and (3) $h_{trunc}^k(\tilde{\mathbf{B}}) = 0$.*

*Proof.* First notice that $(\tilde{\mathbf{B}}^k)_{j\ell} > 0$ if and only if there is at least a directed walk with length $k$ from node $j$ to node $\ell$ in $\mathcal{G}$. We consider the following cases:

- (1) $\implies$ (2): Since $\mathcal{G}$ is a DAG, the diagonal entries of $\tilde{\mathbf{B}}^{k+1}$ are zero. Therefore, it suffices to consider its non-diagonal entries. For a DAG, any directed path in a DAG must be a simple path without loop. If it is not, then we can find a directed path, where a node appears twice, which means that there is a loop in the graph and it is in contradiction with the acyclic property. Thus if $\mathcal{G}$ is a DAG, there will be no path with length larger than $k$, and so the non-diagonal entries of $\tilde{\mathbf{B}}^{k+1}$ are zero. Combining both cases, we have $\tilde{\mathbf{B}}^{k+1} = 0$.

- (1) $\implies$ (3): Since $\mathcal{G}$ is a DAG, the diagonal entries of $\tilde{\mathbf{B}}^i, i = 1, \ldots, d$ are zeros. Therefore, we must have $\operatorname{tr}\left(\sum_{i=1}^k \tilde{\mathbf{B}}^i\right) = 0$.

- (2) $\implies$ (1): Since $\tilde{\mathbf{B}}^{k+1} = 0$, $\tilde{\mathbf{B}}$ is nilpotent and $\mathcal{G}$ must be acyclic.

- (3) $\implies$ (1): It is clear that a directed graph is a DAG if and only if it does not contain simple cycle. Since $\operatorname{tr}\left(\sum_{i=1}^k \tilde{\mathbf{B}}^i\right) = 0$, all path with length smaller than or equal to $k$ must not be a (simple) cycle. If there exists any path with length larger than $k$ that forms a simple cycle, then there exists a simple path that is longer than $k$, which contradicts our assumption that $k$ is the longest simple path. Combining both cases, $\mathcal{G}$ does not contain simple cycle and thus must be acyclic.

Here we proved that (1) and (2) are equivalent; (1) and (3) are equivalent. Thus the three conditions are equivalent under the assumption of the proposition.  □

### A.3  Proof of Proposition 3

We first describe a property of the polynomials of matrices in Lemma 2, which will be used to prove Proposition 3.

**Lemma 2.** *Given $\tilde{\mathbf{B}} \in \mathbb{R}^{d \times d}_{\geqslant 0}$, assume that we have two series $[\alpha_0, \alpha_1, \ldots, \alpha_k] \in \mathbb{R}^{k+1}_{\geqslant 0}$ and $[\beta_0, \beta_1, \ldots, \beta_k] \in \mathbb{R}^{k+1}_{\geqslant 0}$. If we have $\alpha_i \geqslant \beta_i$ for $i = 0, 1, \ldots k$, then the following inequalities hold:*

$$\left\| \sum_{i=0}^{k} \alpha_i \tilde{\mathbf{B}}^i \right\|_F \geqslant \left\| \sum_{i=0}^{k} \beta_i \tilde{\mathbf{B}}^i \right\|_F, \tag{13a}$$

$$\left\| \sum_{i=0}^{k} \alpha_i \tilde{\mathbf{B}}^i \right\|_\infty \geqslant \left\| \sum_{i=0}^{k} \beta_i \tilde{\mathbf{B}}^i \right\|_\infty. \tag{13b}$$

*Proof.* By the assumption that $\tilde{\mathbf{B}} \in \mathbb{R}^{d \times d}_{\geqslant 0}$, we have $\tilde{\mathbf{B}}^i \in \mathbb{R}^{d \times d}_{\geqslant 0}$ for $i = 0, 1, \ldots, k$. Then, using the assumption that $\alpha_i \geqslant \beta_i$ for $i = 0, 1, \ldots k$, straightly we can see that each entry of $\sum_{i=0}^{k} \alpha_i \tilde{\mathbf{B}}^i$ should be no less than the corresponding entry of $\sum_{i=0}^{k} \beta_i \tilde{\mathbf{B}}^i$. Thus (13) must hold. $\qquad\square$

With the lemma above, we now present the proof of Proposition 3.

**Proposition 3.** *Given $\tilde{\mathbf{B}} \in \mathbb{R}^{d \times d}_{\geqslant 0}$, if there exists $k < d$ such that $\|\tilde{\mathbf{B}}^k\|_\infty \leqslant \epsilon < \frac{1}{(k+1)d}$, then we have*

$$0 \leqslant h_{geo}(\tilde{\mathbf{B}}) - h_{trunc}^k(\tilde{\mathbf{B}}) \leqslant \frac{1 - (d\epsilon)^{d/k-1}}{1 - (d\epsilon)^{1/k}} d^{2+1/k} \epsilon^{1+1/k},$$

$$0 \leqslant \|\nabla_{\tilde{\mathbf{B}}} h_{geo}(\tilde{\mathbf{B}}) - \nabla_{\tilde{\mathbf{B}}} h_{trunc}^k(\tilde{\mathbf{B}})\|_F \leqslant (k+1)d\epsilon \|\nabla_{\tilde{\mathbf{B}}} h_{geo}(\tilde{\mathbf{B}})\|_F.$$

*Proof.* Let $[\lambda_1, \lambda_2, \ldots, \lambda_d]$ be the $d$ eigen values of $\tilde{\mathbf{B}}$. We have

$$\max_i |\lambda_i^k| \leqslant \|\tilde{\mathbf{B}}^k\|_F \leqslant d\|\tilde{\mathbf{B}}^k\|_\infty \leqslant d\epsilon, \tag{14}$$

which indicates that

$$\max_i |\lambda_i| \leqslant (d\epsilon)^{1/k}. \tag{15}$$

As all entries of $\tilde{\mathbf{B}}$ are non-negative, for all $j > 1$ we have

$$\text{tr}(\tilde{\mathbf{B}}^{k+j}) = |\text{tr}(\tilde{\mathbf{B}}^{k+j})| = \left| \sum_{i=1}^{d} \lambda_i^{k+j} \right| \leqslant \sum_{i=1}^{d} |\lambda_i^{k+j}| \leqslant d \max_i |\lambda_i^{k+j}| = d(\max_i |\lambda_i|)^{k+j} \leqslant d^{2+j/k} \epsilon^{1+j/k},$$

which implies that

$$h_{geo}(\tilde{\mathbf{B}}) - h_{trunc}^k(\tilde{\mathbf{B}}) = \sum_{j=1}^{d-k} \text{tr}(\tilde{\mathbf{B}}^{k+j}) \leqslant \sum_{j=1}^{d-k} d^{2+j/k} \epsilon^{1+j/k} = \frac{1 - (d\epsilon)^{d/k-1}}{1 - (d\epsilon)^{1/k}} d^{2+1/k} \epsilon^{1+1/k}.$$

This finishes the proof of the first part of the proposition. For the second part, we have

$$\|\nabla_{\tilde{\mathbf{B}}}[h_{\text{geo}}(\tilde{\mathbf{B}}) - h_{\text{trunc}}^k(\tilde{\mathbf{B}})]\|_F$$

$$= \underbrace{\left\|\sum_{j=1}^{d-k}(k+j)\,\mathbf{B}^{k+j-1}\right\|_F \leqslant \left\|\sum_{j=1}^{d-k}(kj+j)\,\mathbf{B}^{k+j-1}\right\|_F}_{kj+j \geqslant k+j \text{ for } j=1,2,\ldots d-k, \text{ and then by Lemma } 2}$$

$$= \underbrace{(k+1)\left\|\tilde{\mathbf{B}}^k\sum_{j=1}^{d-k}j\tilde{\mathbf{B}}^{j-1}\right\|_F \leqslant (k+1)\|\tilde{\mathbf{B}}^k\|_F\left\|\sum_{j=1}^{d-k}j\tilde{\mathbf{B}}^{j-1}\right\|_F}_{\text{Submultiplicativity of Frobenius norm}}$$

$$= \underbrace{(k+1)\|\tilde{\mathbf{B}}^k\|_F\left\|\sum_{j=1}^{d-k}j\tilde{\mathbf{B}}^{j-1} + \sum_{j=d-k+1}^{d}0\tilde{\mathbf{B}}^{j-1}\right\|_F \leqslant (k+1)\|\tilde{\mathbf{B}}^k\|_F\left\|\sum_{j=1}^{d}j\tilde{\mathbf{B}}^{j-1}\right\|_F}_{\text{By Lemma } 2}$$

$$\leqslant \underbrace{(k+1)d\epsilon\left\|\sum_{j=1}^{d}j\tilde{\mathbf{B}}^{j-1}\right\|_F}_{\text{By Eq. (14)}}$$

$$= (k+1)d\epsilon\|\nabla_{\tilde{\mathbf{B}}}h_{\text{geo}}(\tilde{\mathbf{B}})\|_F.$$

$\square$

## A.4 Proof of Proposition 4

**Proposition 4.** *Given any $d \times d$ real matrix $\tilde{\mathbf{B}}$, let $f_i(\tilde{\mathbf{B}}) = \tilde{\mathbf{B}} + \tilde{\mathbf{B}}^2 + \cdots + \tilde{\mathbf{B}}^i$ and $h_i(\tilde{\mathbf{B}}) = \text{tr}(f_i(\tilde{\mathbf{B}}))$. Then we have the following recurrence relations:*

$$f_{i+j}(\tilde{\mathbf{B}}) = f_i(\tilde{\mathbf{B}}) + \tilde{\mathbf{B}}^i f_j(\tilde{\mathbf{B}}),$$
$$\nabla_{\tilde{\mathbf{B}}}h_{i+j}(\tilde{\mathbf{B}}) = \nabla_{\tilde{\mathbf{B}}}h_i(\tilde{\mathbf{B}}) + (\tilde{\mathbf{B}}^i)^\top\nabla_{\tilde{\mathbf{B}}}h_j(\tilde{\mathbf{B}}) + if_j(\tilde{\mathbf{B}})^\top(\tilde{\mathbf{B}}^{i-1})^\top.$$

*Proof.* Substituting $i+j$ for $i$ in the definition of $f_i(\tilde{\mathbf{B}})$, we have

$$f_{i+j}(\tilde{\mathbf{B}}) = \tilde{\mathbf{B}} + \tilde{\mathbf{B}}^2 + \cdots + \tilde{\mathbf{B}}^i + \tilde{\mathbf{B}}^{i+1} + \tilde{\mathbf{B}}^{i+2} + \cdots + \tilde{\mathbf{B}}^{i+j}$$
$$= (\tilde{\mathbf{B}} + \tilde{\mathbf{B}}^2 + \cdots + \tilde{\mathbf{B}}^i) + (\tilde{\mathbf{B}}^{i+1} + \tilde{\mathbf{B}}^{i+1} + \tilde{\mathbf{B}}^{i+2} + \cdots + \tilde{\mathbf{B}}^{i+j})$$
$$= f_i(\tilde{\mathbf{B}}) + \tilde{\mathbf{B}}^i(\tilde{\mathbf{B}} + \tilde{\mathbf{B}}^2 + \cdots + \tilde{\mathbf{B}}^j)$$
$$= f_i(\tilde{\mathbf{B}}) + \tilde{\mathbf{B}}^i f_j(\tilde{\mathbf{B}}).$$

Similarly, to establish the recurrence relation for $\nabla_{\tilde{\mathbf{B}}}h_{i+j}(\tilde{\mathbf{B}})$, we have

$$\nabla_{\tilde{\mathbf{B}}}h_{i+j}(\tilde{\mathbf{B}}) = \nabla_{\tilde{\mathbf{B}}}\text{tr}(f_{i+j}(\tilde{\mathbf{B}}))$$
$$= \nabla_{\tilde{\mathbf{B}}}\text{tr}(f_i(\tilde{\mathbf{B}})) + \nabla_{\tilde{\mathbf{B}}}\text{tr}(\tilde{\mathbf{B}}^i f_j(\tilde{\mathbf{B}}))$$
$$= \nabla_{\tilde{\mathbf{B}}}h_i(\tilde{\mathbf{B}}) + \nabla_{\tilde{\mathbf{B}}}\text{tr}(\tilde{\mathbf{B}}^i f_j(\tilde{\mathbf{B}}))$$
$$= \nabla_{\tilde{\mathbf{B}}}h_i(\tilde{\mathbf{B}}) + (\tilde{\mathbf{B}}^i)^\top\nabla_{\tilde{\mathbf{B}}}\text{tr}(f_j(\tilde{\mathbf{B}})) + f_j(\tilde{\mathbf{B}})^\top\nabla_{\tilde{\mathbf{B}}}\text{tr}(\tilde{\mathbf{B}}^i)$$
$$= \nabla_{\tilde{\mathbf{B}}}h_i(\tilde{\mathbf{B}}) + (\tilde{\mathbf{B}}^i)^\top\nabla_{\tilde{\mathbf{B}}}h_j(\tilde{\mathbf{B}}) + if_j(\tilde{\mathbf{B}})^\top(\tilde{\mathbf{B}}^{i-1})^\top$$

where the fourth equality follows from the product rule. $\square$

## A.5 Proof of Corollary 1

**Corollary 1.** *Given any matrix $d \times d$ real matrix $\tilde{\mathbf{B}}$, let $f_i(\tilde{\mathbf{B}}) = \tilde{\mathbf{B}} + \tilde{\mathbf{B}}^2 + \cdots + \tilde{\mathbf{B}}^i$ and $h_i(\tilde{\mathbf{B}}) = \mathrm{tr}(f_i(\tilde{\mathbf{B}}))$. Then we have have the following recurrence relations:*

$$f_{2i}(\tilde{\mathbf{B}}) = (\mathbb{I} + \tilde{\mathbf{B}}^i) f_i(\tilde{\mathbf{B}}),$$

$$\nabla_{\tilde{\mathbf{B}}} h_{2i}(\tilde{\mathbf{B}}) = \nabla_{\tilde{\mathbf{B}}} h_i(\tilde{\mathbf{B}}) + (\tilde{\mathbf{B}}^i)^\top \nabla_{\tilde{\mathbf{B}}} h_i(\tilde{\mathbf{B}}) + i \left( f_{2i}(\tilde{\mathbf{B}}) - f_i(\tilde{\mathbf{B}}) + \tilde{\mathbf{B}}^i - \tilde{\mathbf{B}}^{2i} \right)^\top.$$

*Proof.* The first equality is straightforwardly obtained by substituting $i = j$ in the first equality of Proposition 4. Similarly, for the second equality, substituting $i = j$ in the second equality of Proposition 4 yields

$$
\begin{aligned}
\nabla_{\tilde{\mathbf{B}}} h_{2i}(\tilde{\mathbf{B}}) &= \nabla_{\tilde{\mathbf{B}}} h_i(\tilde{\mathbf{B}}) + (\tilde{\mathbf{B}}^i)^\top \nabla_{\tilde{\mathbf{B}}} h_i(\tilde{\mathbf{B}}) + i f_i(\tilde{\mathbf{B}})^\top (\tilde{\mathbf{B}}^{i-1})^\top \\
&= \nabla_{\tilde{\mathbf{B}}} h_i(\tilde{\mathbf{B}}) + (\tilde{\mathbf{B}}^i)^\top \nabla_{\tilde{\mathbf{B}}} h_i(\tilde{\mathbf{B}}) + i \left( \tilde{\mathbf{B}}^i + \tilde{\mathbf{B}}^{i+1} + \cdots + \tilde{\mathbf{B}}^{2i-1} \right)^\top \\
&= \nabla_{\tilde{\mathbf{B}}} h_i(\tilde{\mathbf{B}}) + (\tilde{\mathbf{B}}^i)^\top \nabla_{\tilde{\mathbf{B}}} h_i(\tilde{\mathbf{B}}) + i \left( f_{2i}(\tilde{\mathbf{B}}) - f_i(\tilde{\mathbf{B}}) + \tilde{\mathbf{B}}^i - \tilde{\mathbf{B}}^{2i} \right)^\top.
\end{aligned}
$$

$\square$

## A.6 Proof of Proposition 5

**Proposition 5.** *Given $\tilde{\mathbf{B}} \in \mathbb{R}_{\geqslant 0}^{d \times d}$, if there exists $k$ such that $\|\tilde{\mathbf{B}}^k\|_\infty \leqslant \epsilon < 1/d$, then we have*

$$\left\| (\mathbb{I} - \tilde{\mathbf{B}})^{-1} - \sum_{i=0}^{k} \tilde{\mathbf{B}}^k \right\|_F \leqslant d\epsilon \left\| (\mathbb{I} - \tilde{\mathbf{B}})^{-1} \right\|_F.$$

*Proof.* Under the assumption of the proposition, the spectral radius of $\tilde{\mathbf{B}}$ is upper bounded by $(d\epsilon)^{1/k}$ with similar reasoning in (15), which is smaller than one; therefore, we can write $(\mathbb{I} - \tilde{\mathbf{B}})^{-1} = \sum_{j=0}^{\infty} \tilde{\mathbf{B}}^j$. Thus we have

$$
\begin{aligned}
\left\| (\mathbb{I} - \tilde{\mathbf{B}})^{-1} - \sum_{i=0}^{k} \tilde{\mathbf{B}}^k \right\|_F &= \left\| \sum_{j=1}^{\infty} \tilde{\mathbf{B}}^{k+j} \right\|_F = \left\| \tilde{\mathbf{B}}^k \sum_{j=1}^{\infty} \tilde{\mathbf{B}}^j \right\|_F \\
&\leqslant \|\tilde{\mathbf{B}}^k\|_F \left\| \sum_{j=0}^{\infty} \tilde{\mathbf{B}}^j \right\|_F \leqslant d\|\tilde{\mathbf{B}}^k\|_\infty \left\| (\mathbb{I} - \tilde{\mathbf{B}})^{-1} \right\|_F = d\epsilon \left\| (\mathbb{I} - \tilde{\mathbf{B}})^{-1} \right\|_F.
\end{aligned}
$$

$\square$

# B Fast Truncated Matrix Power Iteration

## B.1 Example Implementation

We provide an example implementation of Algorithm 2 in Listing 1.

```python
1  import numpy as np
2
3
4  def h_fast_tmpi(B, eps=1e-6):
5      d = B.shape[0]
6      global old_g
7      global old_B
8      global sec_g
9      if old_g is None:
10         old_g = np.zeros_like(B)
11         old_B = np.zeros_like(B)
12         sec_g = np.zeros_like(B)
13     if old_g.shape[0] != d:
14         old_g = np.zeros_like(B)
15         old_B = np.zeros_like(B)
16         sec_g = np.zeros_like(B)
17     _B = np.copy(B)
18     _g = np.copy(B)
19     _grad = np.eye(d)
20
21     i = 1
22     while i <= d:
23         np.copyto(old_B, _B)
24         np.copyto(old_g, _g)
25         np.matmul(_B, _g, out=_g)
26         np.add(old_g, _g, out=_g)
27
28         np.copyto(sec_g, _grad)
29         np.matmul(_B.T, _grad, out=_grad)
30         np.matmul(_B, _B, out=_B)
31         np.add(_grad, sec_g, out=_grad)
32         np.copyto(sec_g, _g)
33         sec_g -= old_g
34         sec_g += old_B
35         sec_g -= _B
36         sec_g *= i
37
38         _grad += sec_g.T
39
40         if np.max(np.abs(_B)) <= eps:
41             break
42         i *= 2
43
44     return np.trace(_g), _grad
```

Listing 1: An example implementation of Algorithm 2 in Python.

## B.2 Comparison of Time Complexity of Different DAG Constraints

The time complexity for computing the proposed DAG constraints, along with the existing ones, is provided in Table 2. Note that the complexity is described in terms of matrix multiplications, whose efficient algorithms have received much attention in the field of numerical linear algebra in the past decades. Coppersmith–Winograd [12] algorithm and Strassen algorithm [40] are widely used and require $\mathcal{O}(d^{2.376})$ and $\mathcal{O}(d^{2.807})$ operations, respectively.

# C  Additional Experimental Results

## C.1  Other Performance Metrics

The False Discovery Rate (FDR), True Positive Rate (TPR), and False Positive Rate (FPR) are provided in this section, and our TMPI based algorithm often achieves best performance for all these metrics.

Table 2: Complexity of different DAG constraints terms and their gradients in terms of matrix multiplications.

| DAG constraint term (and its gradient) | Complexity |
|---|---|
| Matrix exponential [50][i] | $\mathcal{O}(d)$ |
| Binomial [45] [ii] | $\mathcal{O}(\log d)$ |
| Polynomial [44] | $\mathcal{O}(d)$ |
| NOBEARS [21] | $\mathcal{O}(1)$ |
| Single term | $\mathcal{O}(\log d)$ |
| Geometric | $\mathcal{O}(d)$ |
| Fast geometric | $\mathcal{O}(\log d)$ |
| **TMPI** | $\mathcal{O}(k)$ |
| **Fast TMPI** | $\mathcal{O}(\log k)$ |

[i] More precisely, the complexity of matrix exponential depends on the desired accuracy of the computation, for which a wide variety of algorithms are available [1, 2]. Here we use a fixed complexity since the first $d$ terms of the Taylor expansion are sufficient to form a DAG constraint. [ii] Based on exponentiation by squaring.

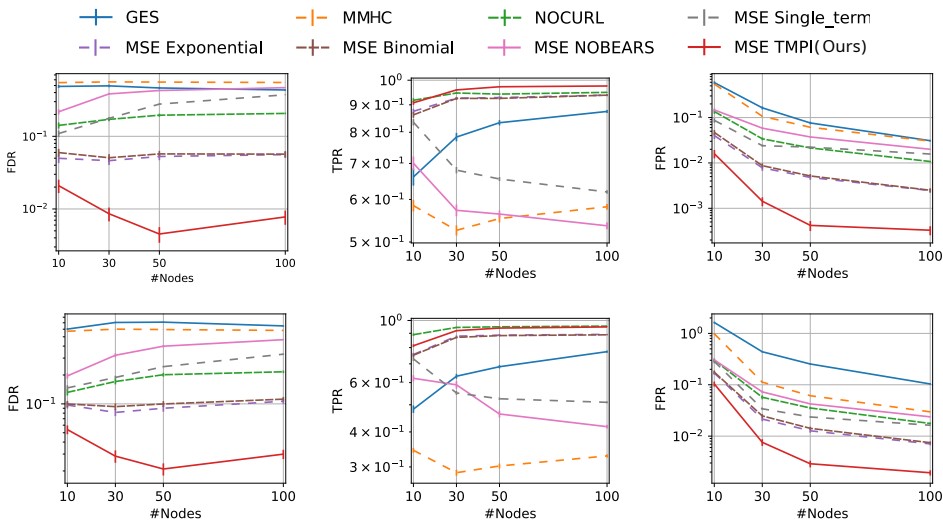

Figure 5: False Discovery Rate (FDR, lower is better), True Positive Rate (TPR, higher is better) and False Positive Rate (FPR, lower is better) of MSE based algorithms on ER2 (**top**) and ER3 (**bottom**) graphs.

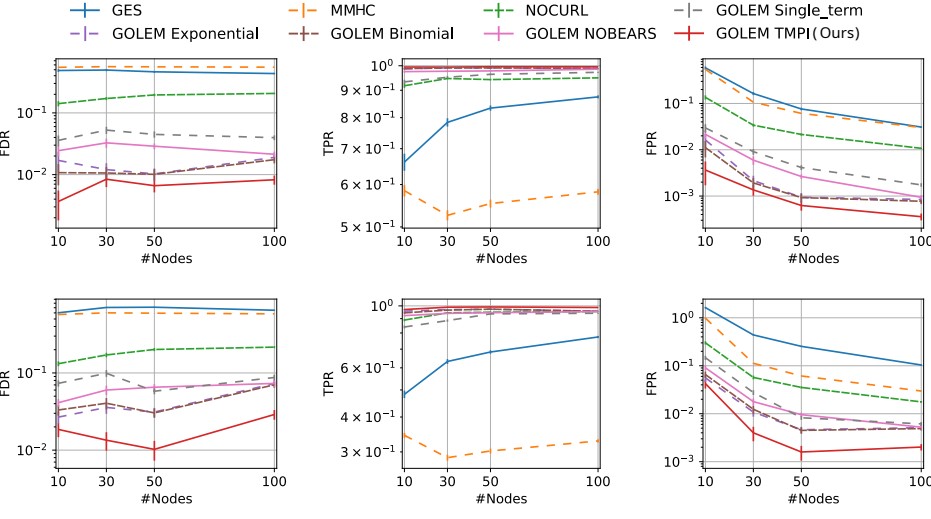

Figure 6: False Discovery Rate (FDR, lower is better), True Positive Rate (TPR, higher is better) and False Positive Rate (FPR, lower is better) of likelihood loss based algorithms on ER2 (**top**) and ER3 (**bottom**) graphs.

## C.2 Experiments on Scale-Free Graphs

We consider the scale-free graph model [3] and the results are shown in Figures 7, where we can see that with different graph structures our algorithm consistently achieves better results.

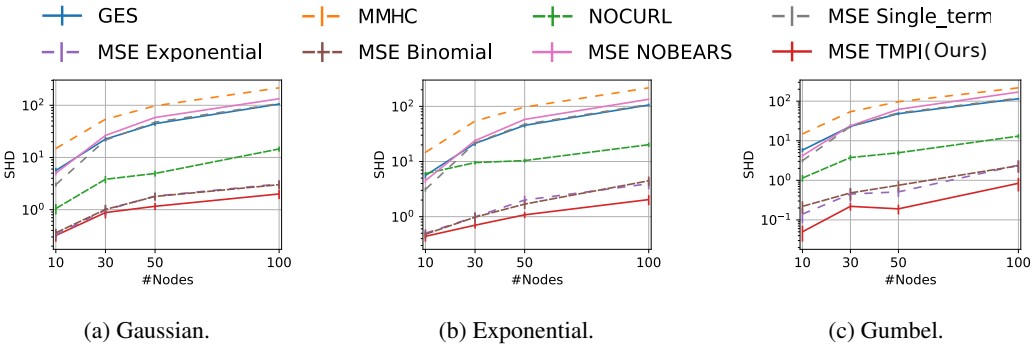

           (a) Gaussian.                   (b) Exponential.                 (c) Gumbel.

Figure 7: SHD of different algorithms on SF2 graphs.

## C.3 Experiments with Smaller Sample Size

We experiment with a smaller sample size, *i.e.*, $n = 200$ samples, on the linear SEM, and the results are shown in Figure 8, where our algorithm outperforms all others in most of the cases.

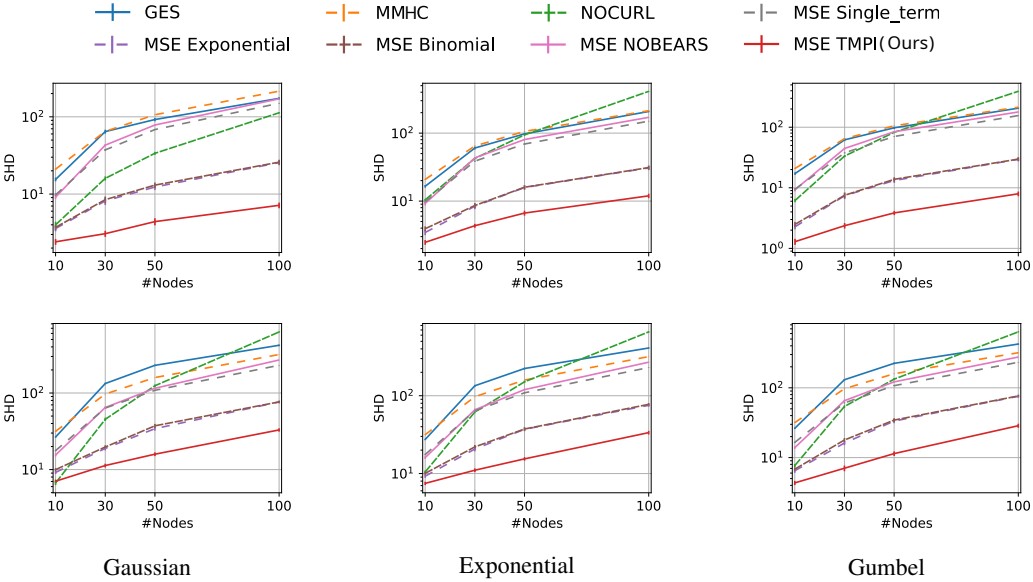

          Gaussian                 Exponential               Gumbel

Figure 8: Results with 200 samples on ER2 (**top**) and ER3 (**bottom**) graphs.

## C.4 Experiments on Larger Graphs

We conduct experiments on ER2 graph with $500$ nodes with Gaussian noises. For these kind of graphs, the GOLEM algorithm is known to perform well [23]. Thus we combine the GOLEM algorithm with different DAG constraints, where for all DAG constraints, the parameters $\eta_1$ and $\eta_2$ for GOLEM algorithm are set to $0.02$ and $5.0$, respectively, and the number of iterations is set to $70000$. The results are shown in Table 3 and our algorithm achieves the best SHD and fastest running time. The running time are measured on a server with V100 GPU.

Table 3: Experiments on 500-node ER2 graph with Gaussian noise. Best results are in bold. All results are averaged from 20 simulations, along with the standard errors.

|  | GOLEM + Exponetial | GOLEM + Binomial | GOLEM + TMPI |
|---|---|---|---|
| SHD ↓ | $45.20 \pm 6.44$ | $46.75 \pm 7.41$ | $\mathbf{14.45 \pm 2.27}$ |
| Running Time(s) ↓ | $882.22 \pm 1.08$ | $795.24 \pm 1.24$ | $\mathbf{670.26 \pm 0.76}$ |

## C.5   Comparison with NOFEARS

Wei et al. [44] and Ng et al. [24] mentioned that the operation $\mathbf{B} \odot \mathbf{B}$ is one source of gradient vanishing. Based on the observation, Wei et al. [44] proposed a KKT condition based local search strategy (named NOFEARS) to avoid gradient vanishing caused by $\mathbf{B} \odot \mathbf{B}$. As shown by Wei et al. [44], to attain best performance the NOFEARS method is applied as a post-processing procedure for NOTEARS[50]. Our method identified a different source of gradient vanishing caused by the small coefficients for higher-order terms in DAG constraints. Thus our work is in parallel with previous work [24, 44], and the local search strategy by Wei et al. [44] can be combined with our technique. We compare the performance of our algorithm and NOFEARS [44] in Figure 9, where our method outperforms the original NOFEARS (NOTEARS+KKTS search) in most cases. The method that combines our method and NOFEARS attains the best performance (*i.e.*, lowest SHD and False Discovery Rate (FDR), highest True Positive Rate (TPR)).

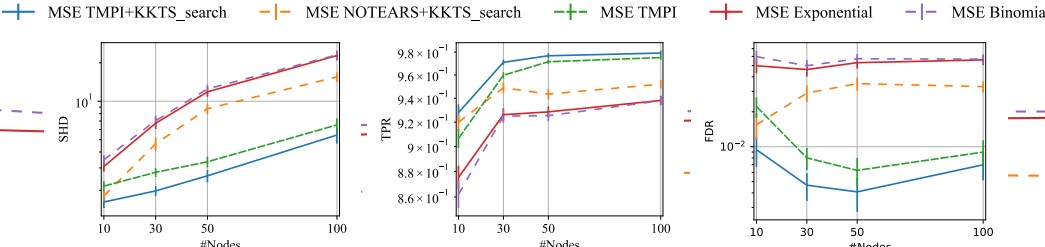

Figure 9: Comparison with KKT based local search strategy [44] on ER2 graphs with linear Gaussian SEMs. The results are averaged from 100 simulations. The NOFEARS method (NOTEARS+KKTS search) achieves better performance tham that of the original NOTEARS (MSE Exponential), but it performs worse than our TMPI method in most cases. The method that combines TMPI and local search (TMPI+KKTS search) achieves the best performance.

## C.6   Experiments on Denser Graphs

We have conducted a simple experiment on ER6 graphs to compare the performance of different polynomial constraints on ER6 graphs (where the expected degree of node is 12). The dataset generating procedure for ER6 graph is the same as Section 4, but the number of samples $n$ is set to 5000. The result is shown in Figure 10, where our TMPI method achieves best performance (*i.e.*, lowest SHD and False Detection Rate (FDR), highest True Positive Rate (TPR)).

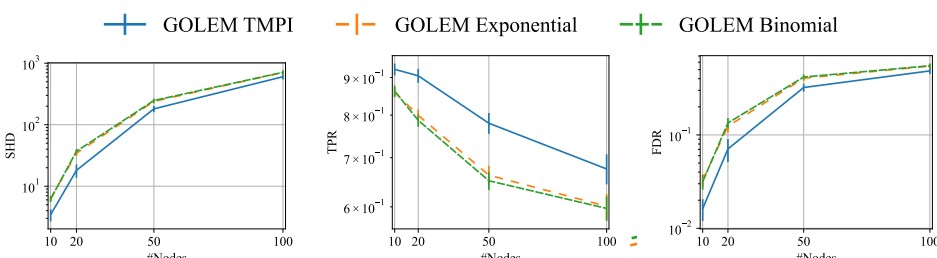

Figure 10: Comparison of different polynomial based DAG constraints on ER6 graphs. All results are averaged from 20 simulations.

## C.7 Additional Ablation Study

We provide an ablation study to illustrate the numeric difference between different $k$ for our DAG constraint, as depicted in Figure 11. One observes that typically the error of our truncated DAG constraints against the geometric series based DAG constraint appears to drop exponentially as $k$ increases. Furthermore, this error may often exceed the machine precision. Therefore, for our TMPI algorithm (*i.e.*, the $\mathcal{O}(k)$ implementation described in Algorithm 1), the value of $k$ is often not large, and thus the TMPI algorithm is usually faster than both the geometric and fast geometric method, as demonstrated by the empirical study in Figure 4. Our fast TMPI algorithm (i.e., the $\mathcal{O}(\log k)$ implementation described in Algorithm 2) further accelerates the procedure and its running time is the shortest in most settings.

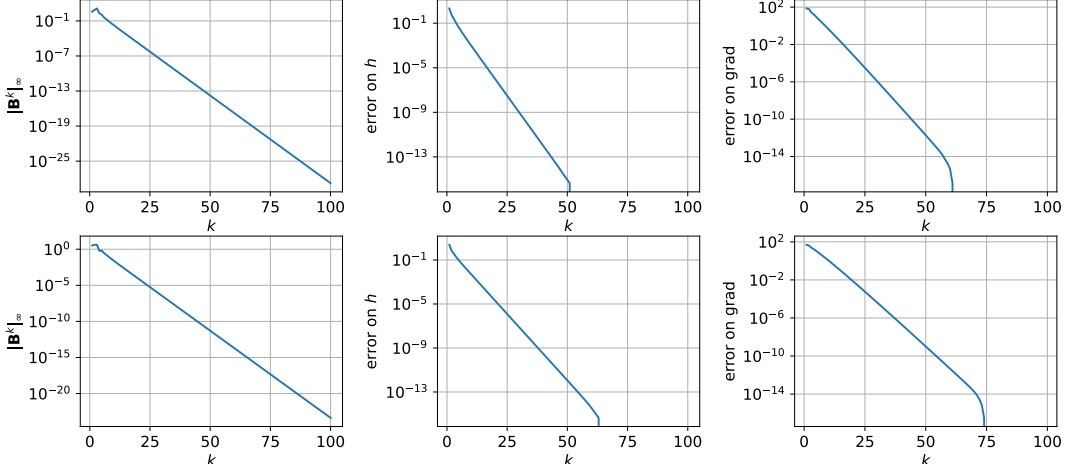

Figure 11: Typical example of $\|\tilde{\mathbf{B}}^k\|_\infty$, error of $h_{\mathrm{trunc}}^k(\tilde{\mathbf{B}})$ against $h_{\mathrm{geo}}(\tilde{\mathbf{B}})$, and error of $\nabla_{\tilde{\mathbf{B}}} h_{\mathrm{trunc}}^k(\tilde{\mathbf{B}})$ against $\nabla_{\tilde{\mathbf{B}}} h_{\mathrm{geo}}(\tilde{\mathbf{B}})$ (in terms of infinity norm). The analysis is based on ER2 (**top**) and ER3 (**bottom**) graphs.