# OpenReview forum: "Truncated Matrix Power Iteration for Differentiable DAG Learning"
_NeurIPS.cc/2022/Conference — NeurIPS 2022 Accept_

### Official Review · Reviewer_iGwL · 2022-06-24

**Rating:** 6
**Confidence:** 4
**Soundness:** 2 fair
**Presentation:** 3 good
**Contribution:** 3 good

**Summary:**

This paper is on learning directed acyclic graphical models using a continuous optimization formulation, where acyclicity is enforced through a matrix polynomial constraint. It is shown that previously proposed polynomial constraints (NOTEARS and similar) suffer from vanishing higher-order terms. A polynomial with unit coefficients is proposed to address this problem, as well as a truncated, lower-order version of this polynomial. The approximation error of the truncated polynomial is bounded. Furthermore, a fast algorithm is presented to compute the new polynomial constraint as well as its gradient. Experiments are performed on synthetic data generated from linear and nonlinear structural equation models, as well as one real dataset. The new polynomial constraint can reduce structural Hamming distance by large factors (i.e. 2-4X) over existing polynomial constraints.

**Questions:**

- Propositions 3 and 5: Which norms are referred to by $\lVert \cdot \rVert_\infty$ and $\lVert \cdot \rVert_2$?
- Proposition 1: Is the reference to [37] in the proof actually to [35]? Is this proposition a corollary of [35, Lemma 1]?
- Figure 1 and related discussion: Clarify what "gradient" and "gradient on higher-order terms" mean.

More for curiosity:
- The SHD of all algorithms, and in particular the strong baselines Exponential and Binomial, are worse by a large factor for the MSE loss in Figure 2 compared to the likelihood loss in Figure 3. The SHDs in Figure 2 are also worse by a similarly large factor compared to previous work with a similar experimental setting. Compare the top row (ER2) in Figure 2 to e.g. the leftmost columns in Figures 1, 5, 6, 7 in [35], which are also for ER2, $n = 1000$, the same values of $d$, and the same noise types. Any idea why?

**Limitations:**

Satisfactory

**Strengths And Weaknesses:**

**Strengths:**
+ Furthers our knowledge about continuous constraints for enforcing acyclicity, by pointing out a shortcoming of previously proposed polynomial constraints and proposing a different polynomial to remedy the problem.
+ The structural Hamming distance improvements in the experiments are large (factors of 2-5).
+ The degree $k$ of the polynomial is adapted to each candidate adjacency matrix $\tilde{B}$.

**Weaknesses:**
I like the paper especially for the first reason above. However, the current version suffers from several technical issues in my view.
- Propositions 3 and 5: There appear to be several problems.
    1. First, which norms are meant by the notation $\lVert \cdot \rVert_\infty$ and $\lVert \cdot \rVert_2$? Previously, $\lVert \cdot \rVert_p$ was defined as the element-wise $p$-norm but I suspect this is no longer the case here.
    2. Based on the proof in Appendix A.3, I will assume that $\lVert \cdot \rVert_p$ now means the induced operator norm, i.e., $\lVert A \rVert_p = \sup_{\lVert x \rVert_p = 1} \lVert Ax \rVert_p$. I am assuming this because in the unnumbered equation above (14), I believe $\lVert A \rVert_\infty$ does bound the spectral radius $\rho(A) = \max_i \lvert \lambda_i(A) \rvert$ from above. However, on the left-hand side of this inequality and in (14), the induced 2-norm $\lVert A \rVert_2$ does not equal $\rho(A)$ for a general non-symmetric matrix (the Wikipedia article on spectral radius has a simple counterexample). The adjacency matrix $\tilde{B}$ and its powers are almost certainly non-symmetric.
    3. Continuing with the assumption that $\lVert A \rVert_2$ is the induced 2-norm, the same error occurs in the second-last line of Appendix A.3. On the other hand, I don't think we can replace $\lVert \cdot \rVert_2$ by $\rho(\cdot)$ as we need the sub-additivity property of norms to go from the third-last line to the second-last line, and the spectral radius is not a norm.
    4. Proposition 5: Again assuming that $\lVert A \rVert_2$ is the induced 2-norm, the same error occurs in the proof in Appendix A.5.
    5. Remark 1: The big-O notation seems to be ignoring the $1 - \epsilon^{1/k}$ in the denominators. I don't think this can be done as $k$ can be fairly large so $\epsilon^{1/k}$ is close to 1. Moreover, $k$ is not a constant as it depends on $\epsilon$.
- Proposition 2: I think the proof of (3) $\to$ (1) is incomplete. Since it is still to be proven that $\mathcal{G}$ is a DAG, one must still address directed paths of length greater than $k$, and probably argue that they cannot exist. Also, the phrases "shorter than $k$" and "less than $k$" should be "less than or equal to."
- Proposition 1: The proof of the "if" part in Appendix A.1 refers to [37], but I could not find any mention of "nilpotent" in the paper by Yuan and Malone (2013). Perhaps the reference is to [35] (Wei et al., 2020), wherein Lemma 1 states that $\mathcal{G}$ is acyclic if and only if $\tilde{B}$ is nilpotent. Indeed, Proposition 1 is a corollary of [35, Lemma 1] since the $\tilde{B}^d = 0$/nilpotent $\to$ DAG direction is exactly the same. For the other direction, [35, Lemma 1] implies that $\tilde{B}$ must be nilpotent, and if it is, then $\tilde{B}^k = 0$ for a $k \leq d$, hence $\tilde{B}^d = 0$ also. If the authors agree with this, then the implication from [35, Lemma 1] needs to be acknowledged.
- Figure 1 and corresponding discussion:
    - It is not clear what gradient is referred to here and what "gradient on higher-order terms" means. I am guessing it is the gradient of the constraint function with respect to $\tilde{B}$ and the "higher-order terms in the gradient" refer to the terms in (7) and similar expressions for the gradient of (3), (4). Perhaps it would be better to show the general form of the gradient and then specialize it to the different cases.
     - The statement "nilpotent properties of DAG result in candidate adjacency matrix with very small spectral radius" needs to be justified mathematically. How this contributes to the vanishing gradient terms seen in Figure 1 should also be better explained.

**Minor comments:**
- Line 43, "these works": Which works does this refer to?
- Eq. (4): Is this missing a $- d$ on the LHS?
- "path": I believe this should be "directed path" to be more precise.
- Eq. (9): Is $p$ the intended exponent rather than $2$?
- Algorithm 2, line 5: I wonder whether the $\tilde{B}$ should be $\tilde{B}_p^{old}$.
- Section 4.1, Results: While the improvements seen here are large, it would be nice to mention for context that factor of $> 2$ improvements upon NOTEARS have been achieved before, e.g. with the KKT search algorithm of [35].
- Figure 4 caption, "our fast": Words missing here. In general, the paper could use careful proofreading for typos, missing/extra words, etc.
- Line 310, $k$ is some constant: As mentioned above, I don't think this is true as it depends on $\epsilon$ and $\tilde{B}$.

---

> ### Author Response · Authors · 2022-08-02
> **Rebuttal**
>
> We greatly appreciate the reviewer's thorough and constructive comments, many of which will help improve the quality of our paper. We attempt to address all the concerns in the following.
>
> ### Proposition 3 and Proposition 5:
>
> Thanks for raising this issue and we apologize for the oversight. We have corrected the two propositions as well as their proofs in the revised version. We have also changed $\lVert \cdot \rVert_2$ to the Frobenius norm $\lVert \cdot \rVert_F$ to avoid any possible confusion, and clarified in Lines 174-175 that we use $\lVert \cdot \rVert_\infty$ to refer to the element-wise infinity norm (i.e., maximum norm).
>
> ### Remark 1
> Thank you for pointing this out which helps improve the clarity of our paper. We initially added the upper bounds with big-O notation in Remark 1 as we intended to improve the interpretability of the bounds in Proposition 3. Following your suggestion and also to avoid any possible confusion, we have removed Remark 1 in the revision.
>
>
> ### Proof of (3) $\rightarrow$ (1) in Proposition 2:
> Thank you for raising this concern. We have updated the proof of Proposition 2 in the revision to explicitly consider the case of "paths of length greater than $k$".
>
>
> ### "Shorter than $k$" and "less than $k$" in Proposition 2:
> We have changed them to "smaller than or equal to $k$" according to your suggestion. Thanks.
>
> ### Worse results than those reported by NOFEARS paper:
>
> An interesting point. Given your comment, we attempted to check the official code of NOFEARS--it seems that the NOFEARS paper adopted a different definition of ER2 as compared to our setup. In particular, the [experiment script](https://github.com/skypea/DAG_No_Fear/blob/master/scripts/run_exp_proposed) appears to indicate that they used expected degree of 2 and 4 for ER2 and ER4 graphs, respectively, which is different from our paper that used expected degree of 4 for ER2 graphs. Therefore, the empirical results of ER2 graphs in our paper should be compared to those of ER4 graphs in the NOFEARS paper, though we are not completely sure. Nonetheless, we have conducted a new experiment to compare with NOFEARS in Appendix C.5. We have observed that our algorithm outperforms original NOFEARS in most cases, and combining our method with NOFEARS always achieves the best performance.
>
> ### Figure 1: "gradient on higher-order terms"
>
> The gradients on different order terms refer to
> $$
> \nabla_{\tilde{\mathbf{B}}}c_itr(\tilde{\mathbf{B}}^i),
> $$
> where, for different algorithms, the coefficients $c_i$ are different. We have clarified this in the caption of Figure 1.
>
> ### Figure 1: Mathematical justification of "nilpotent properties of DAG result in candidate adjacency matrix with very small spectral radius"
>
> In our Proposition 1, we show that $
> h_{single}(\tilde{\mathbf{B}}) = \|\|\tilde{\mathbf{B}}^d\|\|=0
> $ if and only if $
> h_{poly}(\tilde{\mathbf{B}})=0$. To constrain $\tilde{\mathbf{B}}$ to be DAG we encourage $h_{poly}$ to be very small. By the continuity of $h_{single}$ and $h_{poly}$, if $h_{poly}(\tilde{\mathbf{B}}) \leqslant \epsilon$, there must be some $\epsilon'$ such that $h_{single}(\tilde{\mathbf{B}}) < \epsilon'$. Since $h_{single}$ can be used to obtain an upper bound of the spectral radius of $\tilde{\mathbf{B}}$, the spectral radius of the weighted adjacency matrix $\tilde{\mathbf{B}}$ must be small (for candidate solutions that are close to being acyclic). We will add this discussion in Section 3.1 to make it clear in the final version.
>
>
> ### Proposition 1:
> Thanks for spotting this. We have fixed the citation to refer to the NOFEARS paper and also acknowledged the implication from the NOFEARS paper (see Lines 147-148).
>
> ### $k$ is some constant (Line 310)
> We have removed "$k$ is some constant" according to your suggestion.
>
> ### Other minor comments
> We will fix all typos and improve the presentation based on your feedback in the final version. We will also acknowledge that factor of $>2$ improvements upon NOTEARS has been achieved before by NOFEARS.

---

> > ### Comment · Reviewer_iGwL · 2022-08-05
> > **Residual issues**
> >
> > Thanks for the many revisions, especially correcting Propositions 3, 5 and their proofs, and the additional experiment with NOFEARS showing that the best performance is achieved by combining the methods. However, there are some remaining issues.
> > - **Proof of (3) $\to$ (1) in Proposition 2:** Most importantly, I think this proof is still not correct. In particular, if there is a path with length $k+1$ that forms a simple cycle (first and last vertices are the same), then any proper subpath of this cycle is both simple and no longer than $k$. Hence there appears to be no contradiction in this case.
> > - **Proposition 3:** In the second part of the proposition involving the gradients, if one moves $\lVert \nabla_{\tilde{B}} h_{geo}(\tilde{B}) \rVert_F$ to the left-hand side, then this becomes a relative bound on the difference in gradients. Would we not then want the bound $d^2 \epsilon$ to be $< 1$, i.e., $\epsilon < 1/d^2$? On a related note, 1) perhaps it is worth keeping the bound as $(k+1) d \epsilon$, from the final line of the proof, rather than further bounding it by $d^2 \epsilon$. 2) In the first part of the proposition, perhaps we also want $d \epsilon < 1$ for a non-vacuous bound, but this would be subsumed by requiring $d^2 \epsilon < 1$ or $(k+1) d \epsilon < 1$. Note that $\epsilon < 1/d$ is assumed for Proposition 5.
> > - **Proof of Proposition 5:** "spectral radius of $\tilde{B}$ is bounded by $\epsilon^{1/k}$" should be $(d \epsilon)^{1/k}$, but I think the spectral radius is actually irrelevant here. We just need the bound on the Frobenius norm from (15).

---

> > > ### Author Response · Authors · 2022-08-05
> > > **We sincerely thank the reviewer for the further feedbacks and time devoted**
> > >
> > > We sincerely thank the reviewer for the further feedbacks and time devoted, which help improve the clarity of the paper. Please find the response to your comments below. Hope this addresses your concern and please kindly let us know if you have further concerns.
> > >
> > > ### Proof of (3) $\rightarrow$ (1) in Proposition 2:
> > > Thanks for pointing out this out. We have considered this issue and would like to clarify that there are two different definitions of "simple path": (1) a path that repeats no vertex, except that the first and last may be the same vertex, and (2) a path that repeats no vertex. The first definition is used by, e.g., Knuth (1997, p. 363), which we adopt in this work. Therefore, "if there is a path with length $k+1$ that forms a simple cycle (first and last vertices are the same)", then the "simple cycle" leads to a simple path with length $k+1$, which is a contradiction with our assumption that $k$ is the length of longest simple path. We have updated the revision to make our definition of "simple path" clear; see the footnote of Proposition 2 in Section 3.2.
> > >
> > > ### Proposition 3:
> > > We appreciate the insightful suggestion. In light of your suggestion, we have updated the revision to keep the bound as $(k+1)d\epsilon\lVert\nabla_{\tilde{\mathbf{B}}}h_{geo}(\tilde{\mathbf{B}})\rVert_{F}$ instead of $d^2\epsilon\lVert\nabla_{\tilde{\mathbf{B}}}h_{geo}(\tilde{\mathbf{B}})\rVert_{F}$. Following your suggestion regarding non-vacuous bound, we also updated the condition to $\lVert\tilde{\mathbf{B}}^k\rVert_{\infty} \leqslant \epsilon<\frac{1}{(k+1)d}$ and added Remark 1 after Proposition 3.
> > >
> > > ### Proof of Proposition 5:
> > > Thanks for spotting the typo of $\epsilon^{1/k}$, and we have updated the manuscript to change it to $(d\epsilon)^{1/k}$. On the other hand, we added this statement (Lines 511-512) regarding spectral radius because, in order for $(\mathbb{I} -\tilde{\mathbf{B}})^{-1}=\sum_{j=0}^{\infty}\tilde{\mathbf{B}}^{j}$ to hold, the spectral radius of $\tilde{\mathbf{B}}$ must be smaller than $1$. We have made this clear in the revision.
> > >
> > > ### References
> > > Donald E. Knuth, The Art of Computer Programming, volume 1, third edition, 1997.

---

> > > > ### Comment · Reviewer_iGwL · 2022-08-05
> > > > **Proposition 2 again**
> > > >
> > > > If one adopts Knuth's definition of simple path (first and last vertex may be the same), then even the statement of the proposition may be contradictory since "let $k$ be the length of the longest simple path in $\mathcal{G}$" does not rule out any cycles. Similarly, in line 476 of the proof, one cannot conclude that "it can not be a loop." I would advise that in working with DAGs, we use definition 2 (no repeated vertices) so that one does not have to distinguish between a path with no repeated vertices and a "simple path" with identical first and last vertices (i.e. a cycle).

---

> > > > > ### Author Response · Authors · 2022-08-06
> > > > > **We are grateful for the prompt reply and the feedback.**
> > > > >
> > > > > We are grateful for the prompt reply and the feedback. Our responses are provided below. We hope this clarifies your concern and please let us know if there are further concerns.
> > > > >
> > > > > ### "Distinguish between a path with no repeated vertices and a simple path with identical first and last vertices (i.e. a cycle)."
> > > > > Following Knuth (1998, p. 363), we would like to clarify that for both "path" and "simple path", the first and last vertices may be the same, i.e., the usage of these two terms are consistent in our work. We totally agree with the reviewer that using the second definition might be more desirable if we work on purely DAGs throughout the whole paper. At the same time, as is common for continuous optimization approaches for learning DAGs, $\tilde{\mathbf{B}}$ does not necessarily correspond to DAGs throughout the optimization process (e.g., when the penalty coefficient is not large) and might be cyclic. In these cases, it might be desirable if the usage of $k$ and path (as well as simple path) could include the cases with cycles; for instance, see our explanation (regarding the intuition of Proposition 2) in Lines 162-165. Given your concern and to avoid any possible confusion, we will provide a proper definition of "path" and "simple path" in Section 2 in the final version.
> > > > >
> > > > > ### "Proposition 2 may be contradictory since 'let $k$ be the length of the longest simple path in $\mathcal{G}$' does not rule out any cycles. Similarly, in line 476 of the proof, one cannot conclude that 'it can not be a loop'"
> > > > >
> > > > > Thanks for the insightful comment. We would like to clarify that, for the part of the proof the reviewer was referring to, we were trying to prove the cases of (1)$\rightarrow$(2) and (1)$\rightarrow$(3). Therefore, the condition (1) is that $\mathcal{G}$ is acyclic, which rules out the cycles in $\mathcal{G}$. Similarly, in line 476 of the proof, one can conclude that "it can not be a loop" because the acyclicity of $\mathcal{G}$ rules out the loop. In light of your concern, we have restructured the proof of Proposition 2 in the revision into $4$ different cases and tried to make it clearer which conditions are applied in which cases.

---

> > > > > > ### Comment · Reviewer_iGwL · 2022-08-08
> > > > > > **Thanks for clarifying the proof of Proposition 2**
> > > > > >
> > > > > > Thank you for the clarification. I think breaking the proof of Proposition 2 into the 4 cases and rewriting each case as you have done has made it much clearer.
> > > > > >
> > > > > > I think the paper is now in an acceptable form and will thus increase my score, but only to 5 because the initial submission had more technical issues than I would like to see.

---

> > > > > > > ### Author Response · Authors · 2022-08-09
> > > > > > > **Thank you**
> > > > > > >
> > > > > > > We are grateful that the reviewer increased the score. First of all, we are wondering whether there is a way to acknowledge your contribution to the paper (we hope that contributions from high quality reviewers like you should clearly be recognized by the community). If you have any way, please kindly let us know.
> > > > > > >
> > > > > > > At the same time, we are wondering whether you could kindly focus on the updated version, because eventually only the updated version will be published to the community.

---

> > > > > ### Author Response · Authors · 2022-08-07
> > > > > **Thanks**
> > > > >
> > > > > Dear Reviewer iGwL,
> > > > >
> > > > > Once again, we greatly appreciate all your valuable feedbacks and time devoted. We have provided responses and revised the paper in light of your concerns and suggestions. As the deadline for the end of author-reviewer discussion approaches, would you mind checking it and confirming if you have further concerns?
> > > > >
> > > > > Best regards,
> > > > >
> > > > > Authors

---

### Official Review · Reviewer_CoNX · 2022-07-07

**Rating:** 6
**Confidence:** 3
**Soundness:** 3 good
**Presentation:** 3 good
**Contribution:** 3 good

**Summary:**

This paper consider continuous score-based DAG learning optimization problem, which is optimizing a least square objective with continuous constraint that characterizes acyclicity for directed graphs. The paper recognizes the gradient vanishing is a important issue in this optimization problem, and most of literature have this issue since they put small weights on the higher order terms in the constraint. The paper proposes a new constraint to avoid gradient vanishing with a trucation to accelerate and ensure stability. Some experiments are conducted to show the effectiveness.

**Questions:**

- The paper claims that the gradient vanishing is a key challenge for continuous score-based DAG learning, while current literature all put small weights on higher order terms in the constraint, thus share this problem. Based on the paper, Fig 1 only shows TMPI constraint can control the gradient of higher order terms, but it is still not clear why and how gradient vanishing is important. Does it hinder the optimization process? convergence of algorithm? or statistical property of the estimator?

- If the improment of TMPI is on optimization side, it would be interesting to see the runtime comparison among different methods. And it would be better to provide discussion or intuition why TMPI performs much better in terms of graph recovery (SHD) than other methods while they are essentially optimizing the same least square objective under DAG constraint.

-  How is the proposed constraint related to the ones other than NOTEARS in existing work? Does it also resolve the issues recognized by others (not gradient vanishing)? If not, is it obvious how they can be combined?

**Limitations:**

One limitation is discussed right after *Remark 1* and also in *Conclusion*, about all polynomial based DAG constraints, which is when the longest path in the graph is long, the higher order term in constraint matters. It is interesting to see discussion on some particular examples, like Markov chain, especially when the coefficients $B_{jk} > 1$.

**Strengths And Weaknesses:**

# Strength

- Solving continuous score-based DAG learning optimization problem is a topic with significance;
- Lots of experiments are conducted to show the effectiveness of proposed new constraint compared to existing work;
- Smooth presentation and discussion about this thread of continuous DAG learning literature.

# Weakness

- It is unclear why gradient vanishing is important, and comparison with existing work is not detailed enough, see Questions;
- Most experiments are about graph recovery, since this is an optimization paper, runtime comparison with other methods is more interesting.

---

> ### Author Response · Authors · 2022-08-02
> **Rebuttal**
>
> We sincerely thank the reviewer for the helpful feedback and time devoted. Below we give a point-by-point response to the comments.
>
> ### Importance of gradient vanishing and "why TMPI performs much better in terms of graph recovery"
>
> Thanks for asking this question. To enforce DAGness, we have to enforce that every simple path in a graph is not a cycle. For a polynomial based DAG constraint, it then requires order-k polynomial where $k$ is the length of longest simple path. In this case, we have to use higher-order polynomials as DAG constraints. As we shown in Figure 1, the gradients on higher order terms
> $$
> \nabla_{\tilde{\mathbf{B}}}c_itr(\tilde{\mathbf{B}}^i)
> $$
> may converge to zero matrices very quickly due to improper selection of the coefficient $c_i$. In this work, we adopted larger coefficients to resolve this issue, which lead to improvements for the structure recovering performance as compared to the baselines (see Section 4). A possible explanation is that this higher-order information is crucial for DAG learning, and that when small coefficients are placed on them (as in existing works), their resulting gradients will be vanishingly small, and thus the weight updates during the optimization process can hardly utilize this higher-order information. As a consequence, only lower-order information is effectively captured during the optimization process, which may be detrimental to the quality of learned DAG. We have added this explanation in Lines 121-123 of Section 3.1 in the revision.
>
> ### Comparison of running time
> In our new experiment on large scale graphs (in Appendix C.4 of the revised version), we compared the average running time of different polynomial based DAG constraints, and our algorithm runs notably faster than the existing polynomial based DAG constraints. We would like to clarify that our main goal is to obtain DAG solutions with higher qualities. We propose the efficient algorithm because, although naively using the geometric series based DAG constraints can lead to very good results, it can be computationally expensive. In our Section 4.3 we showed that replacing geometric series based DAG constraint with our fast TMPI DAG constraint does not lead to performance loss, but require much less computation.
>
> ### Relation to other works
>
> Two most related works are  Wei et al. (2020) and Ng et al. (2022), which have identified another source of gradient vanishing. In particular, they showed that the gradient of the DAG constraint function is zero for DAG solutions (which thus leads to another "gradient vanishing" issue); therefore, for NOTEARS, one has to increase the DAG penalty coefficient to very large values to obtain approximately DAG solutions. However, this is different from the gradient vanishing issue identified in our work, which focuses on vanishing gradients for higher order terms in the polynomial DAG constraint function. Therefore, it is straightforward to combine our proposed DAG constraint with the method developed by Wei et al. (2020); see the experiment in Appendix C.5.
>
> Another relevant work is the GOLEM method proposed by Ng et al. (2020), which helps resolve the ill-conditioning issue of NOTEARS described above (i.e., one has to increase the DAG penalty coefficient to very large values) by developing a likelihood-based method based on soft constraints. In this case, it is also straightforward to combine our proposed DAG constraint with the GOLEM method; see Eq. (11) and Figure 3. Apart from GOLEM, our proposed DAG constraint can also be easily combined with nonlinear extension of NOTEARS, e.g., DAG-GNN and NOTEARS-MLP; see Section 4.2.
>
> ### References
> D. Wei, T. Gao, and Y. Yu. DAGs with no fears: A closer look at continuous optimization for learning Bayesian networks. In Advances in Neural Information Processing Systems, 2020.
>
> I. Ng, S. Lachapelle, N. R. Ke, S. Lacoste-Julien, and K. Zhang. On the convergence of continuous constrained optimization for structure learning. In International Conference on Artificial Intelligence and Statistics, 2022.
>
> I. Ng, A. Ghassami, and K. Zhang. On the role of sparsity and DAG constraints for learning linear DAGs. In Advances in Neural Information Processing Systems, 2020.

---

> > ### Comment · Reviewer_CoNX · 2022-08-08
> > **Reply to rebuttal**
> >
> > I thank the authors for the helpful clarification and additional experiments. I keep my current score.

---

> > > ### Author Response · Authors · 2022-08-09
> > > **Thanks**
> > >
> > > We are thankful to the reviewer for reading our response and for the acknowledgement. We will incorporate all the suggestions in the final version. Many thanks again.

---

### Official Review · Reviewer_SUap · 2022-07-10

**Rating:** 7
**Confidence:** 2
**Soundness:** 4 excellent
**Presentation:** 4 excellent
**Contribution:** 2 fair

**Summary:**

This paper proposes a new regularization term for DAG estimation, whose role is to enforce that the connectivity matrix is nilpotent.

The contribution basically amounts to depart from state of the art that considers infinite sums of power terms as a regularization to replace them by a finite sum of un-weighted power terms.
This simple change is motivated in a thorough theoretical way and a fast algorithm is proposed. Performance is good.

**Questions:**

* I am not super knowledgeable about DAG, so that I may be missing something trivial, but I don't understand proposition 2 (3). I do understand that \tilde{B}^{k+1}=0, but I don't really understand why we should have tr(\sum_i \tilde{B}^i)=0. Your explanation "Therefore, Proposition 2 suggests that h_{trunc}^k(\tilde{B})=0 is a valid DAG constraint" is not illuminating in this respect. Please elaborate a bit more for non-knowledgeable readers like me.
* In "The full optimization framework", I don't understand why you are picking B\odot B. It's probably obvious but once again, I need a sentence on that.
* I need more details on the nonlinear real case. Could you please detail that better ? Notably the model and how the algorithm is modified.
* L99: "give"-> given
* L138: "attains far more better" far better
* L225: "we are actually find an": looking for
* L306: "while [...] moderate" reads weird because "moderate" is a negative adjective. Replacing "moderate"-> "promising" makes the sentence closer to what you probably mean (basically "although the previous stuff is good, it suffers from some weakness and fix that")
* L314: "nut-shell"->nutshell

**Limitations:**

models able to handle bigger data are mentioned at the end, but as I wrote it would be good to show that the proposed method works fine on moderate but bigger graphs.

**Strengths And Weaknesses:**

* The paper focuses on the very focused problem of estimating a DAG connectivity matrix. This highly focused contribution is a strength in my view: it stays focused and everything about the paper looks good in my view, from the state of the art to the theoretical developments.
* As another strength, I must mention that the whole paper is very factual and written in a neutral tone. For once, people are not pretending they are "reinventing" or "rethinking" something...

* As a weakness, I could mention the limited scale of the experiments. I understand that the paper is mostly theoretical, but I guess that DAG estimation would be an important topic for much larger networks. Maybe going to millions is out of scope, but doing experiments with intermediate scales like thousands would be a desirable feature.

---

> ### Author Response · Authors · 2022-08-02
> **Rebuttal**
>
> We are grateful for the reviewer's effort and the helpful comments on our paper. Please find the response to your questions below.
>
> ### Proposition 2 (3):
> Thanks for asking this question. For a positive adjacency matrix $\tilde{\mathbf{B}}$, the entry at $i^{\text{th}}$ row, $j^{\text{th}}$ column is nonzero if and only there is a path from $i$ to $j$. Thus $tr(\mathbf{B}^k)$ can be used to indicate if there is an length-$k$ loop in the graph. Since all entries in $\tilde{\mathbf{B}}$ are positive, all entries of $\mathbf{B}^k$ are also positive, which implies $\sum_{l=1}^{d}tr(\tilde{\mathbf{B}}^l)=tr(\sum_{l=1}^{d}\tilde{\mathbf{B}}^{l})$. By enforcing $tr(\sum_{l=1}^{d}\tilde{\mathbf{B}}^{l})=0$ we are enforcing there is no loop in the graph since the longest possible loop has length $d$. In our Proposition 2 (3),  we only consider graph with $k$ as longest simple path. By the fact that a directed graph is a DAG if and only if it does not contain any simple cycle, that is all simple paths are not loop. By enforcing $h_{trunc}^k(\tilde{\mathbf{B}})=tr(\sum_{l=1}^{d}\tilde{\mathbf{B}}^{l})=0$  we are enforcing that all simple paths in the graph are not loops.
>
> ### Using $\mathbf{B}\odot \mathbf{B}$
> The original matrix $\mathbf{B}$ may contain negative entries. Therefore, following NOTEARS (and other differentiable DAG learning methods), we adopt this operation (element-wise squaring) to obtain an adjacency matrix that corresponds to the same graph structure but with all positive entries.
>
> ### More explanation about nonlinear cases
> To give a brief summary, the nonlinear extensions of NOTEARS, such as NOTEARS-MLP and DAG-GNN, apply similar DAG constraints in Eqs. (3) and (4) w.r.t. the weights of multilayer perceptrons. These DAG constraints can be straightforwardly replaced by our proposed DAG constraint in Section 3. We will add a section that provides a detailed explanation about the data and model we used.
>
> ### Large Scale Graphs
> Due to limited time, we have added a small experiment on 500 nodes graph shown in Appendix C.4 according to your kind suggestion, where our DAG constraint outperforms other polynomial based DAG constraint in factor of 3.
>
> ### Other minor comments
>
> We will fix all typos and improve the presentation based on your feedback in the final version.

---

### Official Review · Reviewer_XcDH · 2022-07-16

**Rating:** 5
**Confidence:** 3
**Soundness:** 2 fair
**Presentation:** 3 good
**Contribution:** 2 fair

**Summary:**

This is an interesting paper that seeks to provide new non-cyclic constraint formulations for DAG learning. The key novelty here is the employment of finite geometric series DAG constraint to escape from gradient vanishing . The idea of reducing the polynomial constraint to order-k polynomial also provides some very interesting benefits. The experimental comparison with SoTA methods is comprehensive, although limited to the comparison on accuracy. Overall, the work is well done and provides some new methodology to this field.

**Questions:**

1. For the experimental results, besides SHD and SHDC, the number of learnt edges should also be provided.
2. In numerical tests, the methods were tested only on relatively sparse graphs (with <=3 averaged number of edges). However, it is often more challenging to learn the graph structure on denser graphs. Can the authors provide comparison on at least one denser graph case, say, ER6?
3. Besides SHD, can the authors provide a comparison on computational time, to provide evidence on the claimed efficiency?
4. NOFEARS has also claimed to resolve the vanishing gradient issue. Can the authors compare their method with NOFEARS?
5. According to what I read, it seems DAG no curls aim to accelerate the learning, not to improve the accuracy. Moreover, its accuracy depends on the tunable penalty parameter, which is possibly why the authors observed a deteriorated performance here if they didn't optimize that penalty parameter. This fact should be at least pointed out and discussed in the paper.

**Limitations:**

The limitations were discussed adequately.

**Strengths And Weaknesses:**

Strength:
A new type of DAG constraint is proposed, and analysis is provided. Empirical results show that this new constraint helps to improve the learning accuracy, in terms of SHD.

Weakness:
Given that the possibility of using general polynomial constraints was already proposed and proved in NOFEARS, and they have also proposed a new formulation to resolve the vanishing gradient issue, the analysis in this paper seems incremental. One of the main contribution is an efficient algorithm to evaluate the constraint. However, only the comparison on SHD was provided. Therefore, there is not presented evidence that the proposed method is more efficient than the SoTA methods.

---

> ### Author Response · Authors · 2022-08-02
> **Rebuttal**
>
> We greatly appreciate your insightful comments and time devoted to our work. We attempt to address all the concerns as follows.
>
> ### Comparison with NOFEARS
> Thanks for pointing this out. NOFEARS (Wei et. al. 2020) identified a different source of gradient vanishing, and another work (Ng et. al. 2022) also mentioned the same issue. In particular, they showed that the gradient of the DAG constraint function is zero for DAG solutions (which thus leads to another "gradient vanishing" issue); therefore, for NOTEARS, one has to increase the DAG penalty coefficient to very large values to obtain approximately DAG solutions.
> However, this is different from the gradient vanishing issue identified in our work, which focuses on vanishing gradients for higher order terms in the polynomial DAG constraint function.
>  Therefore, it is straightforward to combine our proposed DAG constraint with the method developed by Wei et al. (2020); see the experiment in Appendix C.5, showing that combining our proposed method (DAG constraint) with the method (KKT-based local search) by Wei et al. (2020) indeed leads to the best performance as compared to either of them alone.
>
>
> ### Number of learned edges
> For synthetic data, we report the structural Hamming distance (SHD), true positive rate (TPR), and false discovery rate (FDR), which are the metrics commonly used in the DAG learning literature. Given your suggestion, we will include a summary of the number of edges in the final version. We believe that it will not affect our key observation and conclusion. For real data, we do not include the number of edges because the results of Gran-DAG and Gran-DAG++ are cited from Lachapelle et. al. (2020), in which no number of edges are provided. We will reproduce their results and include the number of edges in the final version. Thanks.
>
> ### Results on ER6 graphs
> Due to limited time, we have conducted a small experiment on ER6 graphs (in Appendix C.6 of the revised version) according to your kind suggestion, and it shows that our DAG constraint outperforms the two other polynomial based DAG constraint. It is notable that the result may not be aligned with the result of "ER6" graphs reported in previous publication  Yu et. al. (2021). We have checked the synthetic dataset provided by Yu et. al. (2021) via [this link](https://drive.google.com/file/d/1O52SlAHPRw_iFW_sAfm_vR3oMnoEb8am/view), and found that their "ER6" graphs correspond to the ER3 graphs in our setup since their "ER6" graphs have expected degree of 6 (and not 12), which are the same as our ER3 graphs. For the ER6 graphs used in our new experiment, the expected degree of a node is set to 12 so that for each node, the expected number of connected edges is 6. To the best of our knowledge, in recent publications about differentiable DAG learning, ER6 graphs (expected degree of 12) have not been considered, and the densest graphs considered seem to be ER4 graphs (expected degree of 8; see the NOTEARS paper).
>
> ### Comparison of running time
> In our new experiment on large scale graphs (in Appendix C.4 of the revised version), we compared the average running time of different polynomial based DAG constraints, and our algorithm runs notably faster than the existing polynomial based DAG constraints. We would like to clarify that our main goal is to obtain DAG solutions with higher qualities. We propose the efficient algorithm because, although naively using the geometric series based DAG constraints can lead to very good results, it can be computationally expensive. In our Section 4.3 we showed that replacing geometric series based DAG constraint with our fast TMPI DAG constraint does not lead to performance loss, but require much less computation.
>
> ### Parameters of NOCURL
> Thanks for the comment. Currently we are using the default hyperparameters of NOCURL. We will try to fine-tune its hyperparameters to obtain better results and provide a detailed discussion in the paper. According to Yu et. al. (2021), with tuned parameter, it would be possible for NOCURL to achieve similar performance as NOTEARS, and our algorithm outperforms NOTEARS drastically (our SHD is often only 1/5-1/2 of theirs).
>
>
> ### References
> Y. Yu, T. Gao, N. Yin, and Q. Ji. DAGs with no curl: An efficient dag structure learning approach. In Proceedings of the 38th International Conference on Machine Learning, 2021.
>
> S. Lachapelle, P. Brouillard, T. Deleu, and S. Lacoste-Julien. Gradient-based neural DAG
> learning. In International Conference on Learning Representations, 2020
>
> D. Wei, T. Gao, and Y. Yu. DAGs with no fears: A closer look at continuous optimization for learning Bayesian networks. In Advances in Neural Information Processing Systems, 2020.
>
> I. Ng, S. Lachapelle, N. R. Ke, S. Lacoste-Julien, and K. Zhang. On the convergence of continuous constrained optimization for structure learning. In International Conference on Artificial Intelligence and Statistics, 2022.

---

> ### Author Response · Authors · 2022-08-08
> **A Kind Request**
>
> Dear Reviewer XcDH,
>
> Once again, we appreciate your time devoted to reviewing this paper. We have provided responses to your comments and an updated submission. Could you please check whether they properly addressed your concern? Your feedback would be appreciated. Please kindly let us know in case there are other concerns—we hope we will have the opportunity to respond to them. Thank you very much.
>
> Authors

---

> ### Author Response · Authors · 2022-08-09
> **A Kind Request for Further Feedback**
>
> Dear Reviewer XcDH,
>
> We sincerely thank you for your feedback and time. We have provided responses to your comments and an updated submission. Since the author-reviewer discussion period will end in 6 hours, we would like to kindly request for further feedback. Could you please check whether they properly addressed your concern? Thank you very much.
>
> Best regards,
>
> Authors

---

### Author Response · Authors · 2022-08-02
**General Response**


We thank the reviewers for their valuable feedbacks and time devoted to our work. Reviewer iGWL concisely conveyed the intuition of our contribution: "Furthers our knowledge about continuous constraints for enforcing acyclicity, by pointing out a shortcoming of previously proposed polynomial constraints and proposing a different polynomial to remedy the problem." All reviewers appreciate the excellent performance of our approach: "This simple change is motivated in a thorough theoretical way and a fast algorithm is proposed. Performance is good." (R SUap), "Empirical results show that this new constraint helps to improve the learning accuracy, in terms of SHD." (R XcDH), "Lots of experiments are conducted to show the effectiveness of proposed new constraint compared to existing work;" (R CoNX), "The structural Hamming distance improvements in the experiments are large (factors of 2-5)." (R iGwL).

Reviewer iGwL raised a concern about the Propositions 3 and 5, and we have addressed it in the revised version as well as the individual response to Reviewer iGwl. Reviewer XcDH raised a concern about the potential overlap with NOFEARS (Wei et. al. 2020), which has been clarified in both the revised version and the individual response. Reviewer SUap raised a concern on the scalability, which is addressed by adding an additional experiments on 500-node graphs.


We have supplied additional experimental results following the reviewers' suggestions and/or requests, and answered all questions (see more details in the individual responses).


### References
D. Wei, T. Gao, and Y. Yu. DAGs with no fears: A closer look at continuous optimization for learning Bayesian networks. In Advances in Neural Information Processing Systems, 2020.

---

### Author Response · Authors · 2022-08-08
**A Kind Request for Further Feedback**

Dear reviewers,

We sincerely thank all of you for the insightful feedback and time devoted to our work. We have provided responses and revised the paper to address your concerns and incorporate your suggestions. Moreover, we have further updated the revision in light of further feedback from Reviewer iGWL. As the deadline for the end of author-reviewer discussion approaches, we would like to kindly request for further feedback. Would you mind checking them and confirming if you have further feedback?

Best regards,

Authors

---

### Meta-Review · Area_Chair_JjGC · 2022-08-28

**Recommendation:** Accept
**Confidence:** Certain

**Metareview:**

During the initial review phase, the reviewers were mostly positive in their opinions of the manuscript, noting that it is well-written, novel, and solves an important problem. However, the reviewers noted a few perceived flaws and technical weaknesses in this manuscript as well. Fortunately, these issues appear to have been positively resolved in a productive discussion during the author response period, particularly with reviewer iGwL.

Following the response period, the reviewers agreed that the proposed resolutions would significantly strengthen the paper and render it suitable for acceptance.

I strongly encourage the authors to make the requested changes in preparing an updated version of their manuscript.

**Award:**

No

---

### Decision · Program_Chairs · 2022-09-14

Accept